# Stochastic Direct Search Methods for Blind Resource Allocation

**Juliette Achddou**                                                 *juliette.achdou@gmail.com*
*Department of Computer Science*
*Università degli Studi di Milano*

**Olivier Cappé**                                                    *olivier.cappe@cnrs.fr*
*Department of Computer Science*
*ENS Paris*

**Aurélien Garivier**                                                *aurelien.garivier@ens-lyon.fr*
*Department of Mathematics*
*ENS Lyon*

**Reviewed on OpenReview:** *https://openreview.net/forum?id=m1OXBLHOdH*

## Abstract

Motivated by programmatic advertising optimization, we consider the task of sequentially allocating budget across a set of resources. At every time step, a feasible allocation is chosen and only a corresponding random return is observed. The goal is to maximize the cumulative expected sum of returns. This is a realistic model for budget allocation across subdivisions of marketing campaigns, with the objective of maximizing the number of conversions. We study direct search (also known as pattern search) methods for linearly constrained and derivative-free optimization in the presence of noise, which apply in particular to sequential budget allocation. These algorithms, which do not rely on hierarchical partitioning of the resource space, are easy to implement; they respect the operational constraints of resource allocation by avoiding evaluation outside of the feasible domain; and they are also compatible with warm start by being (approximate) descent algorithms. However, they have not yet been analyzed from the perspective of cumulative regret. We show that direct search methods achieves finite regret in the deterministic and unconstrained case. In the presence of evaluation noise and linear constraints, we propose a simple extension of direct search that achieves a regret upper-bound of the order of $T^{2/3}$. We also propose an accelerated version of the algorithm, relying on repeated sequential testing, that significantly improves the practical behavior of the approach.

## 1 Introduction

### 1.1 Motivation: Blind Resource Allocation

In the field of programmatic marketing, advertisers are given daily budgets that they are required to entirely distribute across a number of predefined subdivisions of a campaign. Their goal is to maximize some notion of cumulative reward during the lifetime of the campaign, corresponding to the number of clicks or purchases generated by the campaign. The expected reward generated by each subdivision every day is an unknown function of the daily budget allocated to that campaign. We focus on the context in which, when choosing a specific allocation, the advertiser only observes a noisy version of the total reward. The optimization task faced by the advertiser can thus be formalized as a continuous resource allocation problem, under zeroth order and noisy feedback. Indeed, a key operational constraint is the impossibility to directly access higher-order (derivative) information. Furthermore, every noisy evaluation of the objective function has a

cost related to the value of the function, that needs to be accounted for in the performance criterion. The cumulative reward is thus more relevant than alternative traditional measures of performance based, for instance, on the distance to the optimum or the norm of the gradient of the objective function reached after some iterations. Note that the resource allocation task that we consider is different from that in which the resource constraints are cumulative, i.e. where the budget spans the whole period instead of one day or time-step. Cumulative and step-level constraints lead to distinct optimization problems, none of which being a reduction of the other.

The blind resource allocation task may be seen as a specific instance of the more general model of zeroth-order linearly constrained optimization considered in this paper. For resource allocation, we assume that the learner has access to $d + 1 \in \mathbb{N}$ different resources. To keep up with the dominant convention in optimization, we consider a minimization problem for which the costs may be thought of as minus the rewards. At each round $t \in \{1, \dots T\}$, the learner is allowed to choose her level of consumption of each resource, on a continuous scale from 0 to 1. We impose that the consumption levels of all the resources sum to 1 (corresponding to the constraint of spending all the daily budget in the advertising context). The use of resource $i \in \{1, \dots, d+1\}$ to a level of $x_t^{(i)}$ generates an expected marginal cost $w_i(x_t^{(i)})$. Overall, the expected one-step cost of the learner is given by $\sum_{i=1}^{d+1} w_i(x_t^{(i)})$, where the set of all possible consumption levels $(x_t^{(1)}, \dots, x_t^{(d+1)})$ corresponds to the $d$ dimensional simplex. The goal of the learner is to sequentially minimize the expected cumulative cost over $T$ evaluations of the function, having access only to a noisy version of the expected cost associated to the allocation tried at step $t$. Not only are the cost functions $w_1$ to $w_{d+1}$ unknown, but one cannot observe their individual outputs.

## 1.2 Model

While the main application of interest to us is resource allocation, our results are valid for more general linearly constrained optimization problems. We consider a generic optimization domain $\mathcal{D}$ that is a subset of $\mathbb{R}^d$ defined by linear constraints: $\mathcal{D} = \{x \in \mathbb{R}^d, A_I x \leq u\}$ with $m \in \mathbb{N}$, $A_I \in \mathbb{R}^{m \times d}$, and $u \in \mathbb{R}^m$. At each round $t \in \{1, \dots T\}$, the learner selects $x_t \in \mathcal{D}$ and incurs a cost $f(x_t) + \epsilon_t$, where $\epsilon_t$ is assumed to be a centered $\sigma$-subgaussian noise, with $\sigma$ known to the learner.

The goal of the learner is to minimize the cumulative cost over $T$ evaluations of the function, or equivalently to *minimize the cumulative regret*

$$R_T = \sum_{t=1}^{T} f(x_t) - f(x_\star) \, ,$$

denoting by $x_\star = \arg\min_{x \in \mathcal{D}} f(x)$ the optimal allocation. We make the following regularity assumption on $f$.

**Assumption 1.** *$f$ is continuously differentiable, $\beta$-smooth and a-strongly convex on $\mathcal{D}$.*

Note that under Assumption 1, $f$ has a unique minimum and is bounded from below. Assumption 1 is common in online optimization due to the difficulty of controlling the cumulative cost without this assumption.

In programmatic advertising and economics, it is common to observe marginal returns that decrease as the level of a resource increases (following the so-called law of diminishing returns). Assuming convexity of $f$ on $\mathcal{D}$ is therefore reasonable, since Assumption 1 implies that when $f$ has the form $f(x_t^{(1)}, \dots x_t^{(d)}) = \sum_{i=1}^{d} w_i(x_t^{(i)}) + w_{d+1} \left(1 - \sum_{i=1}^{d} x_t^{(i)}\right)$, the marginal cost functions $w_1$ to $w_{d+1}$ are also convex and hence satisfy the law of diminishing return (viewing $-w_i$ as the marginal utility associated to the $i$-th resource).

## 1.3 Related Works

The discrete counterpart of the resource allocation model in which the resources can only be used up to discrete consumption levels, is a celebrated model of operations research with multiple applications. Its properties have first been discussed by Koopman (1953) who proposed the first algorithmic solution for this problem. Koopman's works have further been extended by Gross (1956); Katoh et al. (1979) who propose more efficient algorithms under specific assumptions on the number of resources and the total consumption

budget. The range of applications is wide, including experimental design, load management in an industrial context, computer scheduling and, more recently, the *adwords* problem introduced by Mehta et al. (2007). Recently, Agrawal & Devanur (2015) studied online and offline resource allocation, motivated by the latter task. With the same motivation, Fontaine et al. (2020) focus on the online and continuous version of resource allocation in which the learner accesses the derivatives. The method studied by Fontaine et al. (2020) extends the bisection method in dimension $d > 1$.

The broader problem of derivative-free optimization in noisy environments has been considered by researchers coming from different horizons. A relevant stream of works originates from the bandit community, which considered this task as an extension of the more traditional multi-armed bandit problem (see e.g Auer et al., 2002). The class of $\mathcal{X}$-armed bandits models focuses on the case where a learner can select actions in a generic measurable space and the mean-payoff function is regular. In (Bubeck et al., 2011), for example, the mean payoff function is supposed to be locally Lipschitz with respect to some dissimilarity measure. Bubeck et al. (2011) and Munos (2014) adopt the approach of hierarchical optimization, in which the optimization domain is iteratively partitioned, resulting in finer and finer partitions, that are required to be balanced in some sense. The learner maintains an upper confidence bound of the goal function that is constant on each cell defined by the finest partition. The algorithm proposed by Bubeck et al. (2011) achieves a regret of the order of $\sqrt{T}$ when the learner knows the exact order of the smoothness at the optimum. However, partitioning the domain in a hierarchical and balanced way is relatively easy when the domain is an hypercube, but is a computational problem in itself when the domain has a more complex form. We also mention that knowing the smoothness is considered a challenge most of the time in black-box optimization, so that several methods have been introduced that are adaptive to the smoothness (Locatelli & Carpentier (2018); Valko et al. (2013); Shang et al. (2019)). We mention that concurrently to HOO based on hierarchical partitioning, Agarwal et al. (2011) has also proposed a different strategy for $\mathcal{X}$-armed bandits, but this time convex, with ellipsoid methods, that also result in $O(\sqrt{T})$ regret.

The extension of more traditional first-order optimization methods has also been considered. When the function evaluation is not perturbed by any noise, Nesterov & Spokoiny (2017) consider random gradient descent based on finite differences to estimate the gradient. Flaxman et al. (2004) consider a version of stochastic gradient descent with a one-point estimate of the gradient for the adversarial setting introduced by Zinkevich (2003) in which at each time step, a new goal function is chosen by an adversary, making it impossible to rely on a two-point estimate of the gradient. In this setting, Flaxman et al. (2004) show an adversarial regret bound of the order of $T^{5/6}$. Later on, Hazan & Levy (2014), Hazan & Li (2016), Bubeck et al. (2017) propose new methods for the same setting, but with adversarial convex or strongly-convex functions, showing improved regret bounds, as low as $\sqrt{T}$. In a stochastic setting that is closer to ours, Akhavan et al. (2020); Bach & Perchet (2016) consider a version of stochastic gradient descent with unbiased estimates of the gradient, obtained by finite differences. While they provide an analysis in term of the regret, they focus on a restricted notion of regret that is different from the one considered in this work. The algorithms that they propose rely on a number of samples used to estimate the gradient at each iteration of the gradient descent algorithm. But the regret only accounts for the cost incurred by the iterates of the gradient descent algorithm and ignores the regret incurred by the samples used for the estimation of the gradient. Moreover, the constraints also do not apply to those samples, meaning that the algorithm is allowed to get samples outside of the feasible domain in order to estimate the gradient. The authors prove an upper bound on their version of the regret, which is of the order of $\sqrt{T}$ when Assumption 1 is satisfied. In Section 2.2 below we will see how evaluations outside of the feasible domain can be avoided, using for instance ideas of Bravo et al. (2018), at the price of an increased regret rate.

## 1.4 Contribution

In this paper, we focus instead on a class of simple but mathematically well grounded algorithms known as direct or pattern search methods. Direct search (Kolda et al., 2003) makes use of the well-known fact that if the objective function is continuously differentiable, then at least one of the directions of any *positive spanning set* (a set that spans the space with non-negative coefficients, abbreviated as PSS in what follows) is a descent direction. It explores the space by evaluating the function at new points that are located in a number of predefined search-directions from the current iterate, at a distance from that iterate that varies

with time. The algorithm moves to a new iterate only if this iterate yields a sufficient improvement of the value of the function (there exist other versions of the algorithm where the sufficient decrease condition is replaced with a constraint on the choice of the trial directions). The sample-complexity of such an algorithm has been analyzed in (Vicente, 2013) in the deterministic and unconstrained setting. Lewis & Torczon (2000) study direct search in linearly constrained domains. Handling the constraints in direct search is quite simple, as it consists in testing only the directions in the set of search-directions that are feasible. Gratton et al. (2015) analyzed direct search with random sets of search-directions instead of predefined ones and later extended the analysis to the case of linearly-constrained domains (Gratton et al., 2019). Dzahini (2022) extended their work by analyzing a similar algorithm in the presence of noise, but without constraints. Dzahini (2022) relies on an assumption on the decrease of the noise. In this paper, we will study direct search algorithms that rely on a number of samples at each point to build tight estimates of the function at the trial points, which can be understood as a way to decrease the noise. Dzahini (2022) analyzes some notion of sample complexity of direct search, which only takes the iterates into account rather than the number of function evaluations needed, which is not appropriate in our setting.

Our purpose is to study these methods that are suitable for the blind resource allocation model, i.e. in particular, compatible with zeroth-order feedback, computationally tractable and that do not require to sample points outside of the feasible domain. Besides satisfying the above requirements, these algorithms have the advantage of being approximate descent algorithms with high probability, a guarantee that is useful in practice, allowing, for instance, warm start from previously tested allocations. The adaptation of direct search to the noisy case is achieved by performing enough sampling to ensure that the algorithm moves to a new iterate only if it results in a sufficient improvement, with high probability. We propose two ways of doing so: the first method (termed FDS-Plan) simply computes the number of necessary evaluations ahead of time, whereas the second one, FDS-Seq, uses a sequential testing strategy to interrupt sampling as early as possible. The algorithms are specified in Section 2. An illustration of the behavior of the proposed algorithms can be found in this same section, alongside an illustration of other baseline strategies, which allows for understanding the specifics of direct search. We analyze the cumulative regret of these algorithms in Section 3, providing an upper bound of their regret of the order of $T^{2/3}$ (up to logarithmic factors), when the optimum is in the interior of the feasible domain. A significant technical challenge for the analysis in terms of regret is that, while in traditional analyses of direct-search, the number of rounds is fixed and the analysis proceeds by looking at the distance to the optimum at each round, here, the number of rounds is random (the indexing of the regret is the actual number of function evaluations instead of the number of rounds). We start Section 3 by the simpler case in which there is neither noise nor constraints, showing that in this basic setup the regret of direct search is bounded by a constant.

## 2 Algorithms

### 2.1 Description of the Algorithms

In Algorithm 1 below, we start by describing the most common version of the direct search method used for deterministic and unconstrained optimization. It requires the setting of an initial point $x_0$ and an initial parameter $\alpha_0$. The learner also specifies a PSS $\mathbb{D}$, that is, a set of directions that spans $\mathbb{R}^d$ with non-negative coefficients. At each iteration, the algorithm sequentially tests points at a distance $\alpha_k$ from the current iterate and in the directions defined by $\mathbb{D}$. If none of the test points results in a sufficient decrease of the function's value, the iteration is declared unsuccessful and the trial radius $\alpha_k$ shrinks by a factor $\theta < 1$, otherwise, the iteration is declared successful and the iterate $x_k$ is moved to the first trial point that results in a sufficient improvement. A decrease is considered to be sufficient if it is larger than some predefined forcing function of $\alpha_k$, that we take here to be quadratic, with a coefficient that can be set by the learner.

The analysis can also be adapted to the presence of a growth factor $\phi \geq 1$ by which the trial radius $\alpha_k$ expands at successful iterations. For simplicity, we choose to focus on the case where $\phi = 1$, as this parameter does not modify the regret rates obtained in Section 3. Also note that Algorithm 1 is a descent algorithm with respect to the iterates $x_k$, i.e., the sequence $(f(x_k))_k$ is decreasing.

Choose $x_0 \in \mathbb{R}^d$, $\alpha_0 > 0$, $\theta < 1$, $c > 0$, $\rho(u) = cu^2$ and a PSS $\mathbb{D}$
**for** $k = 0 \dots K$ **do**
    Set *UnsuccessfulSearch* $\leftarrow$ True
    **for** $v \in \mathbb{D}$ **do**
        Evaluate $f(x_k + \alpha_k v)$
        **if** $f(x_k) - f(x_k + \alpha_k v) \geq \rho(\alpha_k)$ **then**
            Set $x_{k+1} \leftarrow x_k + \alpha_k v$ and $\alpha_{k+1} \leftarrow \alpha_k$
            Set *UnsuccessfulSearch* $\leftarrow$ False
            **Break**
        **end**
    **end**
    **if** *UnsuccessfulSearch* **then**
        Set $x_{k+1} \leftarrow x_k$ and $\alpha_{k+1} \leftarrow \theta \alpha_k$
    **end**
**end**

**Algorithm 1:** Direct Search with sufficient decrease

Obviously, different choices of PSS result in different trajectories of the algorithm. Setting $\mathbb{D}$ as the set of $2d$ vectors of the positive and negative coordinate directions results in the algorithm known as coordinate or compass search. Other frequently considered choices include random directions, as in (Gratton et al., 2015; 2019; Dzahini, 2022).

Choose $x_0 \in \mathbb{R}^d$, $\alpha_0 > 0$, $\theta < 1$, $c > 0$, $\rho(u) = cu^2$
**for** $k = 0 \dots K$ **do**
    Set *UnsuccessfulSearch* $\leftarrow$ True
    Select a set of directions $\mathbb{D}_k$
    Set $N_k = \frac{32\sigma^2 \log(2/\delta)}{\rho(\alpha_k)^2}$.
    Estimate $f(x_k)$ by making $N_k$ samples at $x_k$
      and setting $\hat{f}(x_k) = \frac{1}{N_k} \sum_{j=1}^{N_k} f(x_k) + \epsilon_j$
    **for** $v \in \mathbb{D}_k$ *such that* $x_k + \alpha_k v \in \mathcal{D}$ **do**
        Estimate $f(x_k + \alpha_k v)$ by making $N_k$
        samples at $x_k + \alpha_k v$ and setting
        $\hat{f}(x_k + \alpha_k v) = \frac{1}{N_k} \sum_{j=1}^{N_k} f(x_k + \alpha_k v) + \epsilon'_j$
        **if** $\hat{f}(x_k) - \hat{f}(x_k + \alpha_k v) \geq \rho(\alpha_k)$ **then**
            Set $x_{k+1} \leftarrow x_k + \alpha_k v$ and $\alpha_{k+1} \leftarrow \alpha_k$
            Set *UnsuccessfulSearch* $\leftarrow$ False
            **Break**
        **end**
    **end**
    **if** *UnsuccessfulSearch* **then**
        Set $x_{k+1} \leftarrow x_k$ and $\alpha_{k+1} \leftarrow \theta \alpha_k$
    **end**
**end**

**Algorithm 2:** FDS-Plan

Choose $x_0 \in \mathbb{R}^d$, $\alpha_0 > 0$, $\delta > 0$, $\theta < 1$, $c > 0$,
$\rho(u) = cu^2$
Set *UnsuccessfulSearch* $\leftarrow$ True
**for** $k = 0 \dots K$ **do**
    Select a set of directions $\mathbb{D}_k$
    **for** $v$ *in* $\mathbb{D}_k$ *such that* $x_k + \alpha_k v \in \mathcal{D}$ **do**
        **while** *Condition 1 is not satisfied* **do**
            **if** $n_{v,k} \leq n_{0,k}$ **then**
                Sample at $x_k + \alpha_k v$ and update
                $n_{v,k}$ and the empirical mean
                $\hat{f}_{n_{v,k}}(x_k + \alpha_k v)$.
            **end**
            **else**
                Sample at $x_k$ and update $n_{0,k}$ and
                the empirical mean $\hat{f}_{n_{0,k}}(x_k)$
            **end**
        **end**
        **if** $\hat{f}_{n_{0,k}}(x_k) - \hat{f}_{n_{v,k}}(x_k + \alpha_k v) \geq \rho(\alpha_k)$
        **then**
            Set $x_{k+1} \leftarrow x_k + \alpha_k v$, and $\alpha_{k+1} \leftarrow \alpha_k$.
            *UnsuccessfulSearch* $\leftarrow$ False
            **Break.**
        **end**
    **end**
    **if** *UnsuccessfulSearch* **then**
        Set $x_{k+1} \leftarrow x_k$ and $\alpha_{k+1} \leftarrow \theta \alpha_k$
    **end**
**end**

**Algorithm 3:** FDS-Seq

We apply three sorts of modifications to Algorithm 1 in order to adapt it to the more general model introduced in Section 1.2. The first one consists in sampling a trial point only if it is feasible. The second consists in

allowing changes in the set of directions $\mathbb{D}_k$ considered. This is to account for the fact that the change in search-radius at every round impacts the set of admissible directions, denoted $\mathcal{A}_k$. We thus only need to sample directions that span $\mathcal{A}_k$ positively, and not the whole optimization domain $\mathcal{D}$. The third modification consists in introducing estimation stages that allow building reliable estimates of $f$ at the trial points. We propose two ways of doing so, that result in two different algorithms. The first algorithm that we study is a plug-in version of Algorithm 1 in which we replace $f(x_k)$ and $f(x_k + \alpha_k v)$ with their empirical estimates, consisting of means computed from $N_k = \frac{32\sigma^2 \log(2/\delta)}{\rho(\alpha_k)^2}$ samples. This number of samples guarantees that with high probability, the estimation gap is smaller than $\rho(\alpha_k)/4$, which in turn ensures that an iteration is declared successful only when it leads to a decrease of $f(x_k)$ by at least $\rho(\alpha_k)/2$ and that an unsuccessful iteration cannot occur if there exists a direction $v$ in $\mathbb{D}_k$ such that the decrease achieved by moving to $x_k + \alpha_k v$ is larger than $3\rho(\alpha_k)/2$. The resulting algorithm is termed Feasible Direct Search with a planned number of samples (FDS-Plan) and described in Algorithm 2. We also propose a faster algorithm, Feasible Direct Search with Sequential Tests (FDS-Seq) described in Algorithm 3. For any $v \in \mathbb{D}_k$, instead of planning the number of samples at $x_k + \alpha_k v$ ahead of time, it samples at $x_k + \alpha_k v$ and $x_k$ until either

$$
\begin{cases}
\left| \hat{f}_{n_{0,k}}(x_k) - \hat{f}_{n_{v,k}}(x_k + \alpha_k v) - \rho(\alpha_k) \right| \leq \sqrt{2\sigma^2 \log(1/\delta)\left(\frac{1}{n_{0,k}} + \frac{1}{n_{v,k}}\right)}, \\
\text{or } (n_{0,k} \geq N_k \text{ and } n_{v,k} \geq N_k),
\end{cases}
\tag{1}
$$

$n_{0,k}$ and $n_{v,k}$ denoting the number of samples at $x_k$ and $x_k + \alpha_k v$ and $\hat{f}_{n_{0,k}}(x_k)$ and $\hat{f}_{n_{v,k}}(x_k + \alpha_k v)$ the resulting empirical means. Successful and unsuccessful iterations are defined as in FDS-Plan and trigger the same actions.

The sequential stopping rule is designed to achieve early detection of sufficient decrease, but also to detect as early as possible the cases in which the trial point cannot lead to a sufficient decrease. The first test in Condition 1 is a consequence of the fact that the estimation gap at $x_k$ (respectively $x_k + \alpha_k v$) is $\sigma^2/n_{0,k}$ subgaussian (respectively $\sigma^2/n_{v,k}$ subgaussian). The second test of Condition 1 corresponds to a safeguard preventing from waiting too long when the decrease induced by the trial point is very close to the sufficiency threshold. At worst, the number of evaluations needed is the same as in Algorithm 2. Essentially, the sequential stopping rule reduces the number of evaluations needed for each iteration but maintains the desirable property that with high probability, an iteration is declared successful only if it leads to a decrease of at least $\rho(\alpha_k)/2$ and that an iteration cannot be declared unsuccessful if there exists a direction $v$ in $\mathbb{D}_k$ such that the decrease achieved by moving to $x_k + \alpha_k v$ is larger than $3\rho(\alpha_k)/2$.

## 2.2 Illustration

In order to illustrate graphically the behavior of the proposed methods, we show on Figure 1(a) and 1(d) their trajectories in the case where there are 3 resources $(d = 2)$ and the loss functions associated to each resource $i \in \{1, \ldots, d+1\}$ are of the form $w_i(x) = -\tau_i \frac{\log(1+\gamma x)}{\log(1+\gamma)}$ with $\gamma = 2$, $\tau_1 = 1$, $\tau_2 = 0.45$, and $\tau_3 = 0.95$. We set the horizon to $T = 100,000$ and use a Gaussian noise with standard deviation $\sigma = 0.1$, a realistic value for budget allocation problems. In the symmetric representation of Figure 1, the three vertices correspond to the points where one of the resource is fully saturated (equal to 1) and the edges correspond to linear paths along which one of the resources is set to 0. The contour lines of the target function are materialized by orange lines and the location of the minimum is marked by a black cross. The size of each point is a logarithmically growing function of the number of samples made at this point, and its color is a function of the index of the first round at which it has been sampled. Finally, points at which a successful iteration of FDS-Plan (and FDS-Seq) occurred are circled in blue. The parameters of both versions of feasible direct search are $\alpha_0 = 0.2$, $c = 5$ and $\theta = 0.7$, and the initial point corresponds to the allocation $x(0) = (1/3, 1/3, 1/3)$ (center of the simplex). To make the figure more interpretable, we choose to set a fixed $\mathbb{D}$. The set of directions chosen for these algorithms are the 6 directions that support the edges of the simplex (in both directions).

In a first phase, the algorithm proceeds rapidly by testing directions until a sufficient descent direction is found. Afterwards, when the iterates get closer to the minimizer, the search area iteratively shrinks as finding descent directions becomes harder. In the first phase, the trajectory is similar to a descent path that would result from a gradient descent algorithm while the second phase is closer to the behavior of

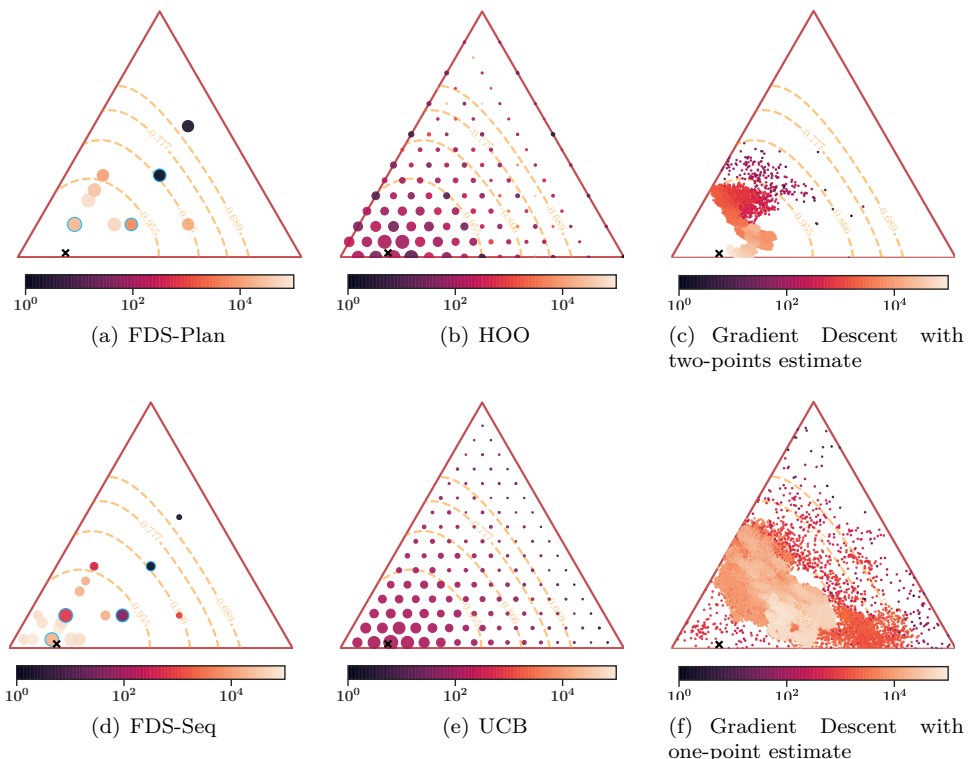

Figure 1: Single trajectories

bandit algorithms based on hierarchical partitions, like HOO (Bubeck et al., 2011). In order to illustrate the differences with such algorithms, we also plot the trajectories of baseline methods, either related to gradient descent or bandits with hierarchical partitioning. Before turning to these other algorithms, it is important to note the difference between FDS-Plan and FDS-Seq. Figure 1(d) shows that FDS-Seq is faster than FDS-Plan, as it spends less time on the first iterations, in which it is easy to determine whether the trial points lead to a sufficient decrease. The FDS-Seq algorithm can thus perform more iterations than FDS-Plan. A common point of these two algorithms that is illustrated on the figure is that they are approximate-descent algorithms with high probability, an interesting quality for practitioners interested in interpretability.

Let us now comment on Figure 1(b), that represents the trajectory of a version of HOO. It is not straightforward to apply algorithms for $\mathcal{X}-$armed bandits like HOO on the simplex, because it implies constructing balanced hierarchical partitions of the simplex. We thus explain our implementation of HOO in Appendix E. On Figure 1(b), we observe that this algorithm explores the partition tree in a way that favors the cells close to the optimum while persistently visiting cells that are clearly far from the minimizer. The behavior of algorithms based on direct search can thus be preferred because it makes warm-start possible, in the sense that prior belief on the location of the minimizer can be used for setting the initial point, which is not possible for HOO. The fact that suboptimal points will keep being sampled until the end of the experiment can also be difficult to accept for practitioners such as advertisers for example.

We also illustrate the behavior of UCB on a discretization of the space, which is an interesting strategy, especially in dimension 2. The discretization used for UCB consists of points arranged in a regular grid of $[0,1]^2$, from which the points lying outside of the feasible domain have been removed. The step parameter of the grid is taken as $T^{-1/4}$ as suggested by Combes & Proutiere (2014). Although simpler than HOO, this algorithm results in similar sampling patterns, as shown on Figure 1(e), and hence shares some of its drawbacks. The performance of UCB is good in dimension 2, as the regret can be proved to be of the order of $\sqrt{T}$ with this choice of step-size (see Combes & Proutiere (2014)) but it will worsen in higher dimension due to the difficulty of simultaneously controlling the distance between grid points and the overall number of

points in the grid. In fact, the optimal step in this case is of the order of $T^{-1/(d+2)}$ and the resulting regret is of the order of $T^{d/d+2}$, which only works in favor of UCB for small values of $d$. Note that while Combes & Proutiere (2014) also provide weaker regret guarantees under more general assumptions, the assumptions required to obtain this order of regret are similar to Assumption 1. They are only less constraining than Assumption 1 in that they require a quadratic upper and lower bound on the function locally near the optimum, whereas Assumption 1 should hold uniformly on the domain.

Lastly, we illustrate the behavior of two methods related to stochastic gradient descent. The first method is related to that proposed by Akhavan et al. (2020). Without constraints, this method would estimate the gradient of the function at $x_t$, by evaluating the function at $y_t^+ = x_t + h_t Z_t$ and $y_t^- = x_t - h_t Z_t$, where $Z_t$ is a random vector of the sphere of radius 1, and use $\frac{f(y_t^+) - f(y_t^-)}{h_t} Z_t$ as an estimation of the gradient. The method in itself does not take constraints into account, but a slight modification results in an algorithm that is feasible in the presence of constraints. This modification consists in performing a homothetic perturbation (Bravo et al., 2018) on the evaluation points $y_t^+$ and $y_t^-$: instead of using these points, the algorithm evaluates the function at $\tilde{y}_t = y_t + h_t/r(c - x_t)$, where $c$ is a point in the interior of the simplex such that $\mathcal{B}(c, r) \subset \mathcal{D}$. We use $\tilde{y}_t^+ = y_t^+ + h_t/r(c - x_t)$ and $\tilde{y}_t^- = y_t^- - h_t/r(c - x_t)$, where $c$ is a point in the interior of the simplex such that $\mathcal{B}(c, r) \subset \mathcal{D}$. This ensures that the evaluation point belongs to the constrained domain, provided that $x_t \in \mathcal{D}$, but adds a bias which is proportional to $h_t$, under suitable regularity assumptions on $f$. To ensure that $x_t \in \mathcal{D}$, we also project the result of the gradient descent step on $\mathcal{D}$. We use an estimation step $h_t$ equal to $(t/2)^{-1/3}$ and a learning rate decreasing as $1/(2.5t)$. This choice is justified by the following reasoning: with this choice of value for the learning rate and $h_t$ set to $t^{-1/4}$, $\sum_{t=1}^{T} f(x_t) - f(x^*)$ would be bounded by $O(T^{1/2})$, thanks to the analysis of Akhavan et al. (2020); but we have to add a term related to the sum of evaluation steps $\sum_{t=1}^{T/2} (f(y_t^+) + f(y_t^-) - 2f(x^*))$ to bound the actual regret, which under suitable assumptions is of order $\sum_{t=1}^{T} h_t$; so that setting $h_t$ to $(t/2)^{-1/3}$ allows to bound both terms by $O(T^{2/3})$. On Figure 1(c), we see one trajectory of this method when the starting iterate is on the center of the simplex. The convergence speed is rather fast at the beginning but the speed is limited by the homothetic perturbation.

The second method is inspired by Flaxman et al. (2004). This paper proposes to use gradient descent with a one-point gradient estimation. In order to evaluate the gradient of the function at $x_t$, the algorithm evaluates the function at $y_t = x_t + h_t Z_t$, where $Z_t$ is a random vector of the sphere of radius 1, and uses $f(y_t)/h_t z_t$ as an estimation of the gradient. As for the previous method, we apply an homothetic perturbation to $y_t$. We use an estimation step $h_t$ equal to $t^{-1/3}$ and a learning rate decreasing as $1/(2.5t)$. The trajectory that we see on Figure 1(f) is not very indicative of the average performance of the algorithm, since this method comes with a very high variance. We see that the algorithm generates a trajectory that roughly gets closer to the minimizer, but that is far from being a descent path because of the poor estimation of the gradient. This method, which has been designed for adversarially evolving objective functions, is clearly not advisable for static objectives with stochastic perturbations.

## 3    Regret Analysis

As discussed in the introduction, the regret criterion takes into account the number $T$ of function evaluations instead of focusing on the number $K$ of iterations, as in the more traditional analysis. In the noiseless case of Algorithm 1, $T$ and $K$ differ by a factor of at most $|\mathbb{D}|+1$ and this is not an issue. However, in the case of Algorithms 2 and 3, the situation is very different as the number of function evaluations per iteration is stochastic and typically increases as the algorithm converges. In this case, it is not possible to predict in advance the evolution of $T$ as a function of $K$ because it depends on the function and starting point. A significant part of the analysis is indeed devoted to quantifying this phenomenon. In practice, it means that in order to comply with a number $T$ of function evaluations set in advance, the algorithms are run without a fixed number of rounds $K$, instead they are run until the number of function evaluations reaches $T$.

In the following, we analyze the proposed algorithms and show that in the constrained and noisy set-up of interest, FDS-Plan and FDS-Seq have a regret of the order of $(\log T)^{2/3} T^{2/3}$ under some further assumptions on $f$ and on the chosen direction set $\mathbb{D}_k$, provided that the optimal point lies in the interior of the feasible do-

main. To provide some intuition on the proofs, we start with the analysis of Algorithm 1 in the unconstrained and deterministic setting, thereby providing the first regret bound of any direct search algorithm.

### 3.1 Warm-up: the Unconstrained and Deterministic Setting

The choice of $\mathbb{D}$ is decisive for the performance of both FDS-Plan and FDS-Seq. In the sequel, we make the following assumption on $\mathbb{D}$.

**Assumption 2.** *The vectors of $\mathbb{D}$ have unit norm and the cosine measure of $\mathbb{D}$ is lower-bounded, i.e, there exists $\kappa > 0$ such that*

$$cm(\mathbb{D}) := \min_{u \in \mathbb{R}^d, u \neq 0} \max_{v \in \mathbb{D}} \frac{u^T v}{\|u\|\|v\|} > \kappa .$$

Assumption 2, common in direct search's literature, guarantees that at each iteration $k$, the cosine similarity between at least one direction in $\mathbb{D}$ and $-\nabla f(x_k)$ is larger than $\kappa$. If $\mathbb{D}$ is a PSS there exists a $\kappa$ satisfying it.

**Theorem 1.** *Under Assumptions 1 and 2, the cumulative regret of Algorithm 1 satisfies*

$$R_T \leq \frac{|\mathbb{D}|+1}{c}\left[ \left( \frac{1}{1-\theta^2} \left( f(x_0) - f(x_\star) + \rho(\alpha_0) \right) \right) \left( \left( 1 + \frac{\eta}{a} \right) \eta + \beta \right) \right.$$
$$\left. + \left( f(x_0) - f(x_\star) \right) \left( \frac{\beta}{a\alpha_0} \|\nabla f(x_0)\| + \beta \right) \right] ,$$

*where $\eta := \frac{\beta}{a} \frac{1}{\kappa\theta}(c + \frac{\beta}{2})$.*

This result shows that under Assumption 1, the asymptotic behavior of the regret of direct search can be compared to that of the more traditional gradient descent algorithm, whose regret is also bounded under this assumption (see Theorem 3.6 of Bubeck et al., 2008).

#### 3.1.1 Elements of Proof

The proof of Theorem 1 combines two well-known properties of direct search and the following lemma holding for any descent algorithm.

**Lemma 1.** *If $f$ satisfies Assumption 1, then $\forall k' > k, \|\nabla f(x_{k'})\| \leq \frac{\beta}{a}\|\nabla f(x_k)\|$.*

In the proof of Theorem 1, Lemma 1 is used in conjunction with the following well-known property (see e.g Vicente, 2013) of direct search.

**Lemma 2.** *If $f$ satisfies Assumptions 1 and 2 and iteration $k$ corresponds to an unsuccessful iteration, then* $\|\nabla f(x_k)\| \leq \frac{1}{\kappa} \left( \frac{\beta}{2}\alpha_k + \frac{\rho(\alpha_k)}{\alpha_k} \right) = \frac{1}{\kappa} \left( \frac{\beta}{2} + c \right) \alpha_k.$

The above lemma follows from the definition of the cosine measure of $\mathbb{D}$, that results in a bound of $v_k^T(-\nabla f(x_k))$, where $v_k$ is the direction in $\mathbb{D}$ maximizing the gap $f(x_k) - f(x_k + \alpha_k v)$, and from the smoothness assumption on $f$. Thanks to Lemma 1, this lemma also means that when iteration $k$ is unsuccessful, we can bound all subsequent gradients by $\frac{\beta}{\kappa a}(\frac{\beta}{2} + c)\alpha_k$. We can deduce that for any $k'$ following the first unsuccessful iteration, $\|\nabla f(x_{k'})\| \leq \eta\alpha_{k'}$, where $\eta := \frac{\beta}{a} \frac{1}{\kappa\theta}(c + \frac{\beta}{2})$. Indeed, if $k'$ is the index of an unsuccessful iteration, Lemma 1 suffices to prove $\|\nabla f(x_{k'})\| \leq \eta\theta\alpha_{k'}$. In contrast, when $k'$ is the index of a successful iteration, one should consider the last unsuccessful iteration $k$. Since $\alpha_{k'} \geq \theta\alpha_k$, there has been at most one reduction of the step-size since iteration $k$ and

$$\|\nabla f(x_{k'})\| \leq \frac{\beta}{\kappa a\theta} \left( \frac{\beta}{2} + c \right) \alpha_{k'} = \eta\alpha_{k'} . \tag{2}$$

The following general argument on direct search is the final key element of the proof of Theorem 1, that links the sum of the squared search-radius to the initial sub-optimality gap $f(x_0) - f(x_\star)$.

**Lemma 3.** *If $f$ satisfies Assumptions 1 and 2,*

$$\sum_{k=0}^{\infty} \rho(\alpha_k) = \sum_{k=0}^{\infty} c\alpha_k^2 \leq \frac{1}{1-\theta^2}(f(x_0) - f(x_\star) + \rho(\alpha_0)) .$$

Lemma 3 can be explained by the fact that $\rho(\alpha_k)$ decreases geometrically by a ratio $\theta^2$ between two successive successful iterations, so that the contribution to the sum of these iterations boils down to multiplying the remainder of the sum by a factor of $\frac{1}{1-\theta^2}$. The sum on successful iterations cannot be too large, because by definition of successful iterations, $f(x_k) - f(x_\star)$ is lower bounded by this sum plus the initial sub-optimality gap $f(x_0) - f(x_\star)$. Bringing Lemma 3 and the bound of Equation 2 together results in a bound on the squared norm of the gradients $\|\nabla f(x_k)\|^2$ after the first unsuccessful iteration. Using the regularity conditions of Assumption 1 suffices to relate the regret to the squared norm of the gradients, which in turn yields Theorem 1. The complete proof can be found in Appendix B.

## 3.2 The Constrained and Noisy Setting

We now turn to the noisy and constrained case described in Section 1.2. We further impose the following assumptions on the domain.

**Assumption 3.** *$\mathcal{D}$ is contained in a ball of radius $b$.*

This assumption, together with assumption 1, implies that $f(x) - f(x_\star)$ is bounded. It also follows from these two assumptions that $\nabla f$ is bounded in norm by a constant, denoted by $B$, on the feasible set $\mathcal{D}$.

While in the unconstrained case, the chosen PSS $\mathbb{D}$ only needed to satisfy Assumption 2, a stronger assumption is required in the presence of linear constraints. Indeed, Assumption 2 was a way to ensure that there was at least one trial direction $v$ in $\mathbb{D}$ satisfying $\frac{-\nabla f(x_k)^T v}{\|v\|\|\nabla f(x_k)\|} \geq \kappa$. This property is not sufficient in the constrained case, because in this case, the directions of interest at iteration $k$ in $\mathbb{D}_k$ are those that are feasible. A problem that might arise for example, is that a sufficient descent direction is not detected even in a situation where $x_k - \alpha_k \nabla f(x_k)$ is feasible, because the set of feasible directions in $\mathbb{D}_k$ does not positively span the feasible region. To avoid such cases, we impose a constraint on $\mathbb{D}_k$ that involves the notion of approximate tangent cones. Approximate tangent cones at a point $x$ are the polar cones of the cones that are generated by the $\alpha$-binding constraints at $x$ as defined by Kolda et al. (2007) (see in particular Figure 2.1 of Kolda et al. (2007) for an illustration of the notion of approximate tangent cones).

Let $a_i^T$ be the $i$-th row of the constraint matrix $A_I$ and let $\mathcal{C}_i = \{y, \text{ s.t } a_i^T y = u_i\}$ denote the sets where the $i$-th constraint are binding. If there exists a point of $\mathcal{C}_i$ at a distance smaller than $\alpha$ from $x$, then the $i$-th constraint is said to be $\alpha$-binding. The indices of $\alpha$-binding constraints at $x$ are denoted $I(x, \alpha) = \{i, \text{ dist}(x, \mathcal{C}_i) \leq \alpha\}$, where dist is induced by the Euclidian distance. We define the approximate normal cone $N(x, \alpha)$ to be the cone generated by the set $\{a_i, \text{ s.t. } i \in I(x, \alpha)\} \cup \{0\}$. The approximate tangent cone $T(x, \alpha)$ is the polar of $N(x, \alpha)$, which means that $T(x, \alpha) = \{v : y^T v \leq 0, \ \forall y \in N(x, \alpha)\}$. Informally $T(x, \alpha)$ is the cone inside of the boundaries generated by the $\alpha$-binding constraints at $x$. We highlight that since the number of constraints $m$ is finite, there can only be a finite number, smaller than $2^m$, of tangent cones. Consequently, Assumption 4 is rather mild.

**Assumption 4.** *For $k \in \{1 \dots K\}$, $\mathbb{D}_k$ contains a set $\mathcal{G}_k$ of positively generating directions of $T(x, \alpha)$ included in $T(x, \alpha)$, for any $x \in \mathcal{D}$ and $\alpha \in \mathbb{R}_+$.*

In the following, we denote by $\mathcal{G}_k$ such a set. Assumption 4 was already necessary in (Kolda et al., 2003) and Lewis & Torczon (2000), while in (Gratton et al., 2019), the descent set at iteration $k$ is assumed to be contained in $T(x, \alpha)$ and to generate it. We explain the purpose of this assumption in the following. While the purpose of Assumption 2 was to ensure that the maximal cosine similarity of a vector in $\mathbb{D}_k$ with $-\nabla f(x_k)$ was bounded away from 0, we focus on a different kind of measure of similarity to $-\nabla f(x_k)$ defined as

$$\begin{cases} \max_{v \in \mathcal{G}_k} \frac{-\nabla f(x_k)^T v}{\|P_{T(x,\alpha)}(-\nabla f(x_k))\|\|v\|} & \text{if } P_{T(x,\alpha)}(-\nabla f(x_k)) \neq 0 , \\ 1 & \text{otherwise.} \end{cases}$$

If $P_{T(x,\alpha)}(-\nabla f(x_k))$ gets close to 0, this measure of similarity to $-\nabla f(x_k)$ does not necessarily become small, although $-\nabla f(x_k)^T v$ is small for any $v$ in $\mathcal{G}_k$. In order to bound this measure of similarity between a vector of $\mathcal{G}_k$ and $-\nabla f(x_k)$ away from 0, we define the following approximate cosine measure:

$$cm_{T(x_k,\alpha_k)}(\mathcal{G}_k) := \inf_{u\in\mathbb{R}^d, P_{T(x_k,\alpha_k)}(u)\neq 0} \max_{v\in\mathcal{G}_k} \frac{u^T v}{\|P_{T(x_k,\alpha_k)}(u)\|\|v\|} \; .$$

As proved by Lewis & Torczon (2000) and recalled by Gratton et al. (2019), if $\mathcal{C}$ is a set of cones $c_j$ that are respectively positively generated from a set of vectors $G(c_j)$, then

$$\lambda(\mathcal{C}) := \min_{c_j\in\mathcal{C}} \left\{ \inf_{u\in\mathbb{R}^d, P_{c_j}(u)\neq 0} \max_{v\in G(c_j)} \frac{u^T v}{\|P_{c_j}(u)\|\|v\|} \right\} > 0 \; ,$$

which guarantees that $cm_{T(x_k,\alpha_k)}(\mathcal{G}_k)$ is bounded by some constant $\kappa_{min} > 0$, under Assumption 4. Under the above assumptions, we can bound the regret of FDS-Plan when the optimum lies in the interior of the feasible set. Note that assuming optimal allocation in the interior of the feasible set is crucial for analysis, but we believe the opposite would not be harmful in practice, as supported by simulations (see Appendix F).

**Theorem 2.** *Under Assumptions 1, 3, and 4, if $x_\star \in int(\mathcal{D})$ and if $|\mathbb{D}_k|$ is bounded by a constant $S_\mathbb{D}$, the cumulative regret $R_T$ of FDS-Plan (respectively FDS-Seq) after the first $T$ evaluations of $f$ satisfies*

$$\mathbb{E}[R_T] = O(\log(T)^{2/3} T^{2/3})$$

*for the choice $\delta = T^{-4/3}$ (respectively $\delta = T^{-10/3}$ for FDS-Seq).*

In the absence of a lower bound, the optimality of such a regret rate is unsure. It is difficult to compare it to other known bounds, as the performance of related algorithms is often not evaluated in the same way. In particular, the performance of the version of stochastic gradient descent proposed by Akhavan et al. (2020) is analyzed with respect to a different notion of regret, $\tilde{R}_T$, that does not take into account the samples needed for the estimation of each gradient. Their analysis yields $\tilde{R}_T = O(\sqrt{T})$. It is important to note that the algorithm by Akhavan et al. (2020) takes advantage of the fact that in the setting of the latter paper, sampling points outside of the feasible domain is possible. When using homothetic perturbation as we did for the illustration in Figure 1(c), the regret of such a method is of the order of $T^{2/3}$, as explained in Section 2.2.

Black-box algorithms such as HOO or StoOO (Bubeck et al., 2011; Munos, 2014) are other possible baselines. When given a balanced hierarchical partition of $\mathcal{X}$, and the smoothness of the function around its optimum, these algorithms would incur a regret of the order of $\sqrt{T}$. The regret rate of FDS-Plan appears to be larger than that of HOO instantiated with the right parameters. However, HOO relies on a partition of the feasible domain that is computationally difficult to achieve with arbitrary linearly constrained domains.

The assumption that $|\mathbb{D}_k|$ be bounded by a constant $S_\mathbb{D}$ is actually not constraining at all, since one way of satisfying Assumption 4 is to set $\mathbb{D}_k$ to the constant set of vectors corresponding to edges of optimization domains, which amounts to $2m(m-1)$ directions, where $m$ is the number of constraints. Depending on the optimization domains, there may be smarter ways of choosing $|\mathbb{D}_k|$ that lead to smaller constants $S_\mathbb{D}$. The motivational case of resource allocation, where the feasible domain is the simplex, is an example of that.

In that case, the above method for choosing $\mathbb{D}_k$ yields $S_\mathbb{D} = 2d(d+1)$ whereas recomputing the direction set at every round can spare us a factor $d$. An intuitive way to understand this is to consider the simplex of dimension $d = 2$. When the iterate is in the interior of the simplex, and the step-size $\alpha_k$ is such that the admissible directions in $T(x_k,\alpha_k)$ form $\mathbb{R}^2$, we only need $d+1 = 3$ vectors (an angle of $2\pi/3$ apart). When $T(x_k,\alpha_k)$ is smaller, then minimal sets $G_k$ are formed by even fewer vectors. An efficient method for recomputing the set of directions at every round is described in Griffin et al. (2008). It is possible to verify that for the simplex, this method provides less than $2d$ directions at each round.

This is particularly important, because the regret bound is proportional to the number of directions contained in $\mathbb{D}_k$ (see the proof of Theorem 2, in Appendix D.2), so that the dependence of the regret with respect to $d$ is linear. For the sake of comparison, the regret of the algorithm by Akhavan et al. (2020) is quadratic in $d$, whereas, when HOO is perfectly parameterized, the dependence on $d$ of the regret of HOO disappears. Recall however, that on arbitrary linearly constrained domain, or even on the simplex in high dimension, HOO might be computationally intractable.

### 3.2.1 Elements of Proof

After some finite number of iterations that depends on $\Delta := \min_{i\in\{1...m\}} \operatorname{dist}(x_\star, \mathcal{C}_i)$, the distance from $x_k$ to the boundaries of $\mathcal{D}$ is smaller than $\Delta/4$ with high probability, thanks to the analysis of Gratton et al. (2019). Waiting for another number of iterations, $\alpha_k$ gets small enough for the approximate tangent cone $T(x_k, \alpha_k)$ to describe the whole space $\mathbb{R}^d$. Then, the trajectory of the algorithm is the same as in the unconstrained setting. In the unconstrained setting, the following elements provide an intuition of why the regret is of the order of $T^{2/3}$. With similar arguments to those of the proof of Theorem 1, i.e Lemmas 1, 2 and 3, it is easy to see that the instantaneous regret incurred at iteration $k$ of the algorithm is proportional to the sum of $\alpha_k^{-2}$ (up to logarithmic factors), whereas it was proportional to $\alpha_k^2$ in the deterministic case: indeed, iteration $k$ now involves $N_k$ times more evaluations than in the noiseless case and $N_k$ is proportional to $\alpha_k^{-4}$ (up to logarithmic factors). Thanks to Lemma 3, we know that $\alpha_k^2$ is summable. Then, thanks to Hölder's inequality applied to the sum of $\alpha_k^{-2}$ written as $(\alpha_k)^{2/3} (\alpha_k)^{-2/3-2}$, the regret is proportional to the total number of evaluations to the power of 2/3, up to logarithmic factors. The complete proof can be found in Appendix D.

## 4 Experiments

In our experiments, we focus on the case in which there are seven resources ($d = 6$), and the loss functions are of the same form as in Section 2.2, and $w_i(x) = -\tau_i \frac{\log(1+\gamma x)}{\log(1+\gamma)}$ with $\gamma = 2$, $\tau_1 = 1$, $\tau_2 = \tau_3 = \tau_4 = 0.75$, $\tau_5 = 0.89$, and $\tau_6 = \tau_7 = 0.95$. On Figure 2, we compare FDS-Seq and FDS-Plan to UCB on a discretization of the space, and gradient descent with an homothetic perturbation. Both methods are explained in Section 2.2. The comparison with HOO is made impossible by the numerical complexity of HOO. We set the horizon to $T = 500,000$ and use a Gaussian noise with standard deviation $\sigma = 0.1$. The set of directions used in FDS-Seq and FDS-Plan are chosen with the method of Griffin et al. (2008). The step parameter of the grid of UCB is set as $T^{-1/(d+2)} = T^{-1/8}$.

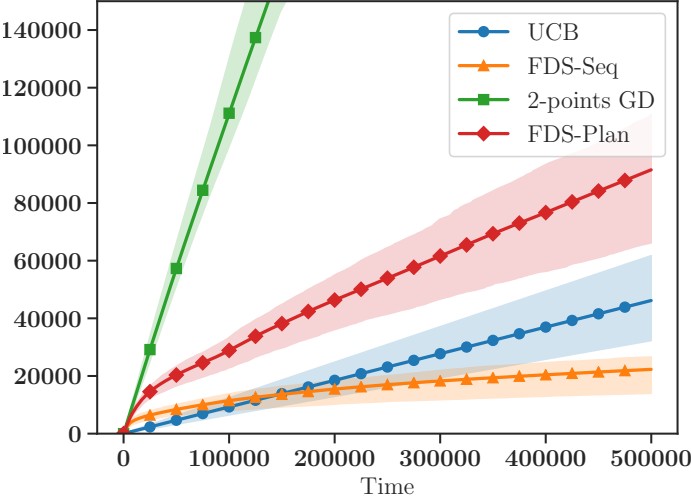

Figure 2: Regret plots of various strategies for resource allocation

The regrets of the algorithms strongly depend on the chosen function. In the case of UCB, the position of the maximizer with respect to the grid that it relies on is important. To alleviate this issue, we plot the mean regret of all the algorithms, when randomly shifting the loss function by a random vector whose coordinates are in $[0.05, 0.05]$. We use 1200 Monte Carlo repetitions. The shaded area represents the region between the first and third quartile.

Clearly, FDS-Seq, by reducing the number of samples needed at the beginning of the run (when moves corresponds to significant drops of the target function), dominates FDS-Plan. The unsatisfying performance of

the gradient descent algorithm can be explained both by the homothetic perturbation that harms the convergence to the optimizer, and by the bad dependence of this algorithm on the dimension. FDS-Seq also clearly outperforms UCB. Note that the performance of UCB will worsen in higher dimension due to the difficulty of simultaneously controlling the distance between grid points and the overall number of points in the grid.

## 5 Conclusion

We have studied extensions of direct search algorithms designed for linearly constrained zeroth-order optimization in the stochastic setting. We have shown that these algorithms, though being fairly simple, suffer a regret of the order of $T^{2/3}$, which is quite satisfactory when compared to other options with comparable implementation cost, like those inspired by finitely-armed bandit algorithms or by gradient descent schemes. There is still a performance gap, in terms of regret rate, when compared to some continuously-armed bandit approaches that are however computationally much more heavy, even in low-dimensional instances of the resource allocation model, such as the one considered in Section 4. We do not believe that the analysis of the algorithms proposed in this paper can be refined so as to obtain the $\sqrt{T}$ regret rate. However, an interesting open question for future work is to know whether this bound could be achieved by other sampling allocation schemes.

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

# Supplementary Material

**Outline.** Appendix A, contains general considerations about the potential repercutions of this (methodological) work. We prove in Appendix B all the results pertaining to the noiseless, unconstrained case. In Appendix C, we provide an additional result for the noisy but unconstrained case. The analysis of direct search in the latter case paves the way for the proof of Theorem 2 whose proof is deferred to Appendix D. Appendix E contains further explanations about simulations in Section 2.2 and Appendix F contains additional experiments.

## A  Broader Impact Statement

This paper is mostly a methodological paper that is unlikely to have a direct societal impact.

However, it explores the idea that direct search algorithms, akin to approximate descent algorithms, can provide explicability in the context of budget allocation for advertising. Advertising practitioners who use these algorithms can explain their actions to their clients by guaranteeing that with high probability, the latter result in an increase of the desired performance indicator. This work is thus part of a collective effort to reach explicability in machine learning, which is crucial as it allows for more transparency.

From an even broader perspective, setting budgets for advertising campaigns is still a manual task in many companies, which could be replaced by algorithms such as those we propose here. Note that this would still leave the task of setting the scope of the campaign (which users to target, on which inventories, etc.) to marketing professionals. It is not clear which impact on employment the automation of budget allocation could have. However, for now, digital marketing is a flourishing sector where employment seems to have increased steadily in the last few years.

## B  Deterministic and unconstrained set-up

### B.1  Preliminary Results

**Lemma 1.** *If $f$ satisfies Assumption 1,*

$$\forall k' > k, \|\nabla f(x_{k'})\| \leq \frac{\beta}{a}\|\nabla f(x_k)\|$$

*Proof.* First observe that because of strong convexity,

$$(\nabla f(x_k) - \nabla f(x_\star))^\top (x_k - x_\star) \geq a\|x_k - x_\star\|^2,$$

and

$$a\|x_k - x^*\|^2 \leq \|\nabla f(x_k)\|\|x_k - x^*\|,$$

which implies that

$$a\|x_k - x^*\| \leq \|\nabla f(x_k)\| . \tag{3}$$

Hence,

$$\begin{aligned}
\|\nabla f(x_{k'})\| &\leq \beta\|x_{k'} - x_\star\| \\
&\leq \frac{\beta}{\sqrt{a}}\sqrt{f(x_{k'}) - f(x_\star)} \leq \frac{\beta}{\sqrt{a}}\sqrt{f(x_k) - f(x_\star)} \\
&\leq \frac{\beta}{\sqrt{a}}\sqrt{\nabla f(x_k)^\top (x_k - x^*)} \leq \frac{\beta}{\sqrt{a}}\sqrt{\|\nabla f(x_k)\|\|(x_k - x^*)\|} \\
&\leq \frac{\beta}{a}\nabla\|f(x_k)\| ,
\end{aligned}$$

where the first inequality comes from the smoothness (in fact $\|\nabla f(x_k)\| = \|\nabla f(x_k)\| - \|\nabla f(x^*)\| \leq \beta \|x_k - x_\star\|$), the second inequality is a result of the strong convexity, the third one ensues from the fact that the algorithm is a descent algorithm, the fourth one arises as a resut of convexity and the fifth one comes from the strong convexity property of Equation 3.

$\square$

**Lemma 2.** *If $f$ satisfies Assumption 1 and iteration $k$ corresponds to an unsuccessful iteration then*

$$\|\nabla f(x_k)\| \leq \frac{1}{\kappa} \left( \frac{\beta}{2} \alpha_k + \frac{\rho(\alpha_k)}{\alpha_k} \right) = \frac{1}{\kappa} \left( \frac{\beta}{2} + c \right) \alpha_k .$$

This lemma is already well-known (see e.g. Vicente, 2013), we only prove it here for completeness.

*Proof.* Since $cm(\mathbb{D}_k) := \min_{u \in \mathbb{R}^d, u \neq 0} \max_{v \in \mathbb{D}_k} \frac{u^\top v}{\|u\|\|v\|} > \kappa$, there exists $v \in \mathbb{D}_k$ such that

$$-\nabla f(x_k)^\top v \geq \kappa \|\nabla f(x_k)\| .$$

Since the iteration is an unsuccessful iteration, we have $f(x_k) - f(x_k + \alpha_k v) \leq \rho(\alpha_k) = c\alpha_k^2$. Then

$$\kappa \alpha_k \|\nabla f(x_k)\| - \rho(\alpha_k) \leq -\nabla f(x_k)^\top v + f(x_k + \alpha_k v) - f(x_k)$$

$$\leq \int_0^{\alpha_k} \nabla f(x_k + uv)^\top v - \nabla f(x_k)^\top v \, du$$

$$\leq \int_0^{\alpha_k} \|\nabla f(x_k + uv) - \nabla f(x_k)\| \|v\| \, du$$

$$\leq \beta \int_0^{\alpha_k} u \, du \leq \frac{\beta}{2} \alpha_k^2 ,$$

which yields $\|\nabla f(x_k)\| \leq \frac{1}{\kappa} \left( \frac{\beta}{2} \alpha_k + \frac{\rho(\alpha_k)}{\alpha_k} \right) = \frac{1}{\kappa} \left( \frac{\beta}{2} \alpha_k + c\alpha_k \right).$ $\square$

**Lemma 3.**

$$\sum_{k=0}^{\infty} \rho(\alpha_k) \leq \frac{1}{1 - \theta^2} (f(x_1) - f(x_\star) + \rho(\alpha_0)) .$$

This lemma is also a common element of the analysis of direct search algorithms (see e.g. Gratton et al., 2019), we only prove it here for completeness.

We assume that there are infinitely many successful iterations as it is trivial to adapt the argument otherwise. Let $k_i$ be the index of the $i$th successful iteration ($i \geq 1$). Define $k_0 = -1$ and $\alpha_{-1} = \alpha_0$ for convenience. Let us rewrite $\sum_{k=0}^{\infty} \rho(\alpha_k)$ as $\sum_{i=0}^{\infty} \sum_{k=k_i+1}^{k_{i+1}} \rho(\alpha_k)$ and study first $\sum_{k=k_i+1}^{k_{i+1}} \rho(\alpha_k)$. Thanks to the definition of the update on a successful iteration and on unsuccessful iterations,

$$\sum_{k=k_i+1}^{k_{i+1}} \rho(\alpha_k) = \sum_{k=k_i+1}^{k_{i+1}} \rho(\theta^i \alpha_{k_i}) = \sum_{k=k_i+1}^{k_{i+1}} c\theta^{2i} \rho(\alpha_{k_i}) \leq \frac{1}{1 - \theta^2} \rho(\alpha_{k_i}) .$$

Since on successes $\rho(\alpha_{k_i}) \leq f(x_{k_i}) - f(x_{k_i} + \alpha_{k_i}) = f(x_{k_i}) - f(x_{k_{i+1}})$,

$$\sum_{i=1}^{\infty} \rho(\alpha_{k_i}) \leq f(x_1) - f(x_\star) .$$

Hence

$$\sum_{k=0}^{\infty} \rho(\alpha_k) \leq \frac{1}{1 - \theta^2} (f(x_1) - f(x_\star) + \rho(\alpha_0)) .$$

**Lemma 4.** *The index $k_f$ of the first unsuccessful iteration satisfies:*

$$k_f \leq \frac{f(x_0) - f(x_*)}{\rho(\alpha_0)} .$$

Another version of this lemma is due to Gratton et al. (2015).

*Proof.* Before the first unsuccessful iteration, $\alpha_k = \alpha_0$. So by definition of a successful iteration, $\forall k$, such that $0 < k \leq k_f$

$$f(x_{k-1}) - f(x_k) \geq \rho(\alpha_0) .$$

By summing,

$$f(x_{k_0}) - f(x_{k_f}) \geq k_f \rho(\alpha_0).$$

The left hand-side of this inequality is upper-bounded by $f(x_0) - f(x_*)$, which suffices to conclude the proof. $\square$

## B.2 Regret Bound

In this section, we prove in Theorem 3 below a result involving the regret at iteration $K$ of the algorithm instead of the regret after $T$ function evaluations. As $T \leq K(S_{\mathbb{D}} + 1)$, Theorem 3 directly implies Theorem 1.

Consider

$$\tilde{R}_K = \sum_{k=0}^{K} \left( f(x_k) - f(x_\star) + \sum_{v \in \mathbb{D}_k} f(x_k + \alpha_k v) - f(x_\star) \right)$$

which is an upper bound of the cumulative regret suffered by the algorithm at iteration $K$, since it accounts for all directions in $\mathbb{D}_k$ at each round $k$, while not necessarily all of them will be tested. $\tilde{R}_K$ can be bounded as follows.

**Theorem 3.** *Under Assumptions 1 and 2,*

$$\tilde{R}_K \leq (S_{\mathbb{D}} + 1) \Big[ \left( \frac{1}{c} \left( \frac{1}{1 - \theta^2} (f(x_0) - f(x_\star) + \rho(\alpha_0)) \right) \right) \left( \left( 1 + \frac{\eta}{a} \right) \eta + \beta \right)$$

$$+ \frac{f(x_0) - f(x_\star)}{\rho(\alpha_0)} \left( \frac{\beta}{a} \|\nabla f(x_0)\| \alpha_0 + \beta \alpha_0^2 \right) \Big] ,$$

*where $\eta := \frac{\beta}{a} \frac{1}{\kappa\theta} (c + \frac{\beta}{2})$.*

*Proof.* We decompose the regret as

$$\tilde{R}_K = \sum_{k=0}^{K} \left( f(x_k) - f(x_\star) + \sum_{v \in \mathbb{D}} f(x_k + \alpha_k v) - f(x_\star) \right)$$

$$\leq \sum_{k=0}^{K} \left( f(x_k) - f(x_\star) + \sum_{v \in \mathbb{D}} f(x_k + \alpha_k v) - f(x_k) + f(x_k) - f(x_\star) \right)$$

$$\leq \sum_{k=0}^{K} ((|\mathbb{D}|+1)(f(x_k) - f(x_\star))) + \sum_{k=0}^{K} \left( \sum_{v \in \mathbb{D}} f(x_k + \alpha_k v) - f(x_k) \right)$$

$$\leq \sum_{k=0}^{k_f} ((|\mathbb{D}|+1)(f(x_k) - f(x_\star))) + \sum_{k=0}^{k_f} \left( \sum_{v \in \mathbb{D}} f(x_k + \alpha_k v) - f(x_k) \right)$$

$$+ \sum_{k=k_f}^{K} ((|\mathbb{D}|+1)(f(x_k) - f(x_\star))) + \sum_{k=k_f}^{K} \left( \sum_{v \in \mathbb{D}} f(x_k + \alpha_k v) - f(x_k) \right), \tag{4}$$

where $k_f$ is the iteration of the first unsuccessful iteration. The third inequality provides a decomposition of the regret in a first term that involves the suboptimality of the iterate, and a second term that involves the difference between values of $f$ at the iterate and at the trial points. As is usual for direct search algorithm, the behavior of the algorithm before the first unsuccessful iteration has to be studied separately, which explains the use of the decomposition of the fourth inequality. We bound the regret due to the rounds preceding $k_f$ by:

**Lemma 5.** *The regret due to the rounds preceding $k_f$ is bounded by*

$$\sum_{k=0}^{k_f} ((|\mathbb{D}|+1)(f(x_k) - f(x_\star))) + \sum_{k=0}^{k_f} \left( \sum_{v \in \mathbb{D}} f(x_k + \alpha_k v) - f(x_k) \right) \le C_1,$$

*where we denote by $k_f$ is the index of the first unsuccessful iteration and by*
$C_1 = \frac{f(x_0) - f(x_\star)}{c\alpha_0^2} \left( (f(x_0) - f(x_\star) + \frac{\beta}{a}\|\nabla f(x_0)\|\alpha_0 + \beta\alpha_0^2 \right).$

*Proof.* As until $k_f$, $f(x_k) \le f(x_0)$ and $\alpha_k = \alpha_0$, it holds that

$$\sum_{k=0}^{k_f} ((|\mathbb{D}|+1)(f(x_k) - f(x_\star))) + \sum_{k=0}^{k_f} \left( \sum_{v \in \mathbb{D}} f(x_k + \alpha_k v) - f(x_k) \right)$$

$$\le \sum_{k=0}^{k_f} ((|\mathbb{D}|+1)(f(x_0) - f(x_\star))) + \sum_{k=0}^{k_f} \left( \sum_{v \in \mathbb{D}} \|\nabla f(x_k)\|\alpha_k + \beta\alpha_k^2 \right)$$

$$\le k_f ((|\mathbb{D}|+1)(f(x_0) - f(x_\star))) + (|\mathbb{D}|+1)k_f \left( \frac{\beta}{a}\|\nabla f(x_0)\|\alpha_0 + \beta\alpha_0^2 \right)$$

$$\le \frac{f(x_0) - f(x_\star)}{c\alpha_0^2} \left( (|\mathbb{D}|+1)(f(x_0) - f(x_\star) + \frac{\beta}{a}\|\nabla f(x_0)\|\alpha_0 + \beta\alpha_0^2) \right)$$

$$= C_1 \, ,$$

where Lemma 1 is used for the second inequality and the third inequality comes from Lemma 4. The first inequality results from the following property of convex and $\beta$-smooth functions: $f(y) - f(x) \le \nabla f(y)^T (x - y) \le \|\nabla f(x)^T (x - y)\| + \beta\|x - y\|^2$, applied to $x_k + \alpha_k v$ and $x_k$. □

**Lemma 6.** *After $k_f$,*

$$\sum_{k=k_f}^{K} f(x_k + \alpha_k v) - f(x_k) \le C_2$$

*where $k_f$ is the index of the first unsuccessful iteration and*
$C_2 = \frac{1}{c}(\eta + \beta)\left( \frac{1}{1-\theta^2}(f(x_0) - f(x_\star)) + \rho(\alpha_0) \right) \, .$

*Proof.* We take $k > k_f$. Using the property of convex and $\beta$-smooth functions that $f(y) - f(x) \le \|\nabla f(x)^T (x - y)\| + \beta\|x - y\|^2$, applied to $x_k + \alpha_k v$ and $x_k$, as in Lemma 5, we get

$$f(x_k + \alpha_k v) - f(x_k) \le \alpha_k \|\nabla f(x_k)\| + \beta\alpha_k^2 \, .$$

We note that if $k$ is the index of an unsuccessful iteration,

$$\|\nabla f(x_k)\| \le \frac{1}{\kappa}\left( c + \frac{\beta}{2} \right)\alpha_k = \frac{1}{L_1'}\alpha_k \, ,$$

by Lemma 2, if $\alpha_k \le 1$. If $k$ is the index of a successful iteration, we can come back to the last unsuccessful iteration $k'$, since

$$\|\nabla f(x_k)\| \le \frac{\beta}{a}\|\nabla f(x_{k'})\| \le \frac{\beta}{a}\frac{1}{\kappa}\left( c + \frac{\beta}{2} \right)\alpha_{k'} \le \frac{\beta}{a}\frac{1}{\kappa}\left( c + \frac{\beta}{2} \right)\frac{\alpha_k}{\theta} \, .$$

where the first inequality comes from Lemma 1, and the third from the fact that $\alpha_k \geq \theta \alpha_{k'}$. Hence for any $k > k_f$,

$$\|\nabla f(x_k)\| \leq \eta \alpha_k \ ,$$

and

$$f(x_k + \alpha_k v) - f(x_k) \leq (\eta + \beta)\alpha_k \ .$$

Hence

$$\sum_{k=0}^{K} f(x_k + \alpha_k v) - f(x_k) \leq \sum_{k=0}^{K} (\eta + \beta)\, \alpha_k^2 \ .$$

Consequently,

$$\sum_{k=0}^{K} f(x_k + \alpha_k v) - f(x_k) \leq (\eta + \beta)\frac{1}{c}\left(\sum_{k=0}^{\infty} \rho(\alpha_k)\right)$$

$$\leq \frac{1}{c}(\eta + \beta)\left(\frac{1}{1 - \theta^2}(f(x_0) - f(x_\star) + \rho(\alpha_0))\right) \ .$$

$\square$

**Lemma 7.**

$$\sum_{k=k_f}^{K} (f(x_k) - f(x_\star)) \leq \frac{1}{ac}\eta^2\left(\frac{1}{1 - \theta^2}(f(x_1) - f(x_\star) + \rho(\alpha_0))\right) := C_3$$

*Proof.* Take $k > k_f$. Thanks to the convexity of $f$,

$$f(x_k) - f(x_\star) \leq \nabla f(x_k)^\top (x_k - x_\star)$$

$$\leq \frac{1}{a}\|\nabla f(x_k)\|^2 \ ,$$

where the second inequality stems from Equation 3, which itself come from strong convexity.

As in the proof of Lemma 6 we have for any $k > k_f$,

$$\|\nabla f(x_k)\| \leq \eta \alpha_k \ ,$$

so that for any $k > k_f$,

$$f(x_k) - f(x_\star) \leq \frac{1}{a}(\eta)^2\left(\frac{\alpha_k}{\theta}\right)^2 \ .$$

Thanks to Lemma 3, we have $\sum_{k=0}^{\infty} \alpha_k^2 \leq \left(\frac{1}{c}\frac{1}{1-\theta^2}(f(x_1) - f(x_\star) + \rho(\alpha_0))\right)$.

Eventually,

$$f(x_k) - f(x_\star) \leq \frac{1}{a}\eta^2\left(\frac{1}{c}\frac{1}{1 - \theta^2}(f(x_1) - f(x_\star) + \rho(\alpha_0))\right) = C_3 \ .$$

$\square$

Using the regret decomposition of Equation 4 together with Lemmas 4, 6, and 7 completes the proof of Theorem 3. $\square$

# C   Noisy and unconstrained set-up

Before considering the constrained setting, we analyze the algorithms described in Section 2 (Algorithms 2 and 3) when there are no constraints, that is, $\mathcal{D} = \mathbb{R}^d$.

## C.1   Presentation of the main Result

**Theorem 4.** *Assume that $f$ is lower bounded and upper bounded on $\mathbb{R}^d$, so that there exits $U$, $f(x) - f(x_\star) \leq U$, $\forall x \in \mathbb{R}^d$. Also assume that the region $\mathcal{X} = \{x \in \mathbb{R}^d : f(x) < f(x_0)\}$ is convex and that $f$ is $a$-strongly convex and $\beta$-smooth on $\mathcal{X}$. Let $R_T$ be the cumulative regret on the $T$ first evaluations of $f$ made by FDS-Plan. Set $\delta = T^{-4/3}$. Then*

$$\mathbb{E}[R_T] = O(\log(T)^{2/3} T^{2/3})$$

This regret bound is also valid for FDS-Seq under the same Assumptions, with $\delta = T^{-10/7}/2$. In the following, we give a proof of the regret bound for FDS-Plan. Note that Sections C.3 and C.2 refer to FDS-Plan, and Section C.4 deals with FDS-Seq.

The regularity assumption in Theorem 4 requires that $f$ is bounded and satisfies a local version of Assumption 1. The initial point $x_0$ should not be chosen too far from $x^*$, nor should $\alpha_0$ be too large. This assumption is not unreasonable, since for every $x_0$ and $\alpha_0$, it is naturally satisfied by bounded and strictly convex functions in $\mathscr{C}^2$ for some choice of $a$ and $\beta$. We stress that under the alternative assumption 1, the same kind of regret bound could still be proved, but with a smaller choice of $\delta$, resulting in higher confidence bonuses and the multiplication of the regret by some constant factor. Indeed, in this case, estimating $f$ incorrectly at each round can lead to a trajectory that always deviates from $x^*$, which is highly detrimental to the regret rate; meanwhile, under the assumption required by Theorem 4, $f$ is bounded by $U$, so that deviating from $x^*$ contributes to the regret by at most $UT$.

In the following, we will use the following additional notation.

**Notation.**   We define $v_k$ to be

$$v_k := \begin{cases} \arg\max_{v \in \mathbb{D}_k} f(x_k) - f(x_k - \alpha_k v) & \text{if iteration } k \text{ is unsuccessful} \\ \text{the chosen direction} & \text{otherwise.} \end{cases}$$

## C.2   Intermediate results

**Lemma 8.** *We call $\mathcal{E}_k$ the event*

$$\mathcal{E}_k = \{|f(x_k + \alpha_k v) - \hat{f}(x_k + \alpha_k v)| \leq c/4(\alpha_k)^2\}, \ \forall v \in \mathbb{D}_k \cup \{0\}\} \ .$$

*The probability of $\mathcal{E}_k$ is lower bounded by*

$$\mathbb{P}\left(\mathcal{E}_k | \mathcal{F}_{k-1}\right) \geq 1 - \delta(S_{\mathbb{D}} + 1)$$

*where $\mathcal{F}_{k-1}$ is the $\sigma$-field representing the history.*

*Proof.* Let $v \in \mathbb{D} \cup \{0\}$  $f(x_k + \alpha_k v) - \hat{f}(x_k + \alpha_k v) = \sum_{i=1}^{N_k} \epsilon_j$ with $\epsilon_j$ independent Gaussian variables with variance $\sigma^2$ and we have

$$\left| \sum_{i=1}^{N_j} \epsilon_j \right| \leq \sqrt{2\sigma^2 \frac{\log(2/\delta)}{N_j}} \leq \sqrt{\frac{2\sigma^2 \log(2/\delta)}{32\sigma^2 \log(2/\delta)/\rho(\alpha_k)^2}} \leq \frac{\rho(\alpha_k)}{4}$$

with probability $1 - \delta$, when knowing $N_k$. By a union bound, $\mathbb{P}\left(\mathcal{E}_k | \mathcal{F}_{k-1}\right) \geq 1 - \delta(|\mathbb{D}|+1)$ where $\mathcal{F}_{k-1}$ is the $\sigma$-field representing the history. $\square$

The following lemma characterizes unsuccessful iterations and successes when $\mathcal{E}_k$ occurs.

**Lemma 9.** *On $\mathcal{E}_k$, if $k$ is an unsuccessful iteration then $f(x_k) - f(x_k + \alpha_k v_k) \leq 3c/2(\alpha_k)^2$ and if $k$ is a successful iteration then $f(x_k) - f(x_k + \alpha_k v_k) \geq c/2(\alpha_k)^2$.*

Lemma 9 implies that if $\mathcal{E}_k$ occurs for all $k$, then each iteration of the algorithm results in a descent.

**Lemma 10.** *On $\cap_{k \leq K} \mathcal{E}_k$, the algorithm is a descent algorithm. In particular, $x_k \in \mathcal{X}$, $\forall k \in \{1 \dots K\}$.*

**Lemma 11.** *If $f$ satisfies the assumptions of Theorem 4 and on $\cap_{k \leq K} \mathcal{E}_k$ then,*

$$\forall k' > k, \|\nabla f(x_{k'})\| \leq \frac{\beta}{a} \|\nabla f(x_k)\| \ .$$

*Proof.* The proof of Lemma 1 applies verbatim thanks to Lemma 10. □

**Lemma 12.** *If $f$ satisfies the assumptions of Theorem 4 and the iteration $k$ corresponds to an unsuccessful iteration then on $\mathcal{E}_k$,*

$$\|\nabla f(x_k)\| \leq \frac{1}{\kappa}\left(\frac{\beta}{2}\alpha_k + \frac{3\rho(\alpha_k)}{2\alpha_k}\right) = \frac{1}{2\kappa}\left(\beta\alpha_k + 3c\alpha_k^2\right) \ .$$

*Proof.* We reproduce the proof of Lemma 2 by using Lemma 9.

Since $cm(\mathbb{D}) := \min_{v \in \mathbb{R}^d} \max_{v \in \mathbb{D}} \frac{v^T v}{\|v\|\|v\|} > \kappa$, there exists $v \in \mathbb{D}$ such that

$$-f(x_k)^\top v \geq \kappa \|\nabla f(x_k)\|.$$

Since the iteration is an unsuccessful iteration, we have $f(x_k) - f(x_k + \alpha_k v) \leq \frac{3}{2}\rho(\alpha_k) = \frac{3}{2}c\alpha_k^2$, thanks to Lemma 9. Then

$$\kappa\alpha_k\|\nabla f(x_k)\| - \rho(\alpha_k) \leq \frac{\beta}{2}\alpha_k^2 \ ,$$

exactly as in the proof of Lemma 2, which yields $\|\nabla f(x_k)\| \leq \frac{1}{\kappa}\left(\frac{\beta}{2}\alpha_k + \frac{3}{2}\frac{\rho(\alpha_k)}{\alpha_k}\right) = \frac{1}{\kappa}\left(\frac{\beta}{2}\alpha_k + \frac{3}{2}c\alpha_k\right)$. □

**Lemma 13.** *If $f$ satisfies the assumptions of Theorem 4 and on $\cap_{k \leq K} \mathcal{E}_k$,*

$$\sum_{k=0}^{K} \rho(\alpha_k) \leq \frac{2}{1 - \theta^2}(f(x_1) - f(x_\star) + \rho(\alpha_0)) \ .$$

Assume that $\cap_{k \leq K} \mathcal{E}_k$ holds. Let $k_i$ be the index of the $i$-th successful iteration ($i \geq 1$). Define $k_0 = -1$ and $\alpha_{-1} = \alpha_0$, and $\alpha_k = 0$, $\forall k > K$ for convenience. Define $K_I$ the number of successes until $K$. We rewrite $\sum_{k=0}^{K_I} \rho(\alpha_k)$ as $\sum_{i=0}^{K_I} \sum_{k=k_i+1}^{k_{i+1}} \rho(\alpha_k)$ and study first $\sum_{k=k_i+1}^{k_{i+1}} \rho(\alpha_k)$.

Thanks to the definition of the update on a successful iteration and on unsuccessful iterations,

$$\sum_{k=k_i+1}^{k_{i+1}} \rho(\alpha_k) \leq \frac{1}{1 - \theta^2}\rho(\alpha_{k_i}) \ .$$

exactly as in the proof of Lemma 3. Since on successes,

$$\frac{1}{2}\rho(\alpha_{k_i}) \leq f(x_{k_i}) - f(x_{k_i} + \alpha_{k_i}) = f(x_{k_i}) - f(x_{k_{i+1}}) \ ,$$

we have

$$\frac{1}{2}\sum_{i=1}^{K_I} \rho(\alpha_{k_i}) \leq f(x_1) - f(x_\star).$$

Hence

$$\sum_{k=0}^{K_I} \rho(\alpha_k) \leq \frac{2}{1 - \theta^2}(f(x_1) - f(x_\star) + \rho(\alpha_0)) \ .$$

**Lemma 14.** *On $\cap_{k \leq K} \mathcal{E}_k$, the first iteration that results in an unsuccessful iteration occurs at round $k_f$, satisfying:*

$$k_f \leq 2 \frac{f(x_0) - f(x_*)}{\rho(\alpha_0)} .$$

Before the first unsuccessful iteration, $\alpha_k = \alpha_0$. So by Lemma 9, $\forall 0 < k \leq k_f$

$$f(x_{k-1}) - f(x_k) \geq \frac{1}{2} \rho(\alpha_0) .$$

By summing,

$$f(x_{k_0}) - f(x_{k_f}) \geq \frac{1}{2} k_f \rho(\alpha_0) .$$

The left hand-side of this inequality is upper-bounded by $f(x_0) - f(x_*)$, which suffices to conclude the proof.

**Lemma 15.** *If $f$ satisfies the assumptions of Theorem 4 and on $\cap_{k \leq K} \mathcal{E}_k$, for any $k$ after the first unsuccessful iteration,*

$$\|\nabla f(x_k)\| \leq \eta_2 \alpha_k,$$

*where we denote by $\eta_2 = \frac{\beta}{2 a \kappa \theta}(3c + \beta)$.*

*Proof.* For unsuccessful iterations,

$$\|\nabla f(x_k)\| \leq \frac{1}{2\kappa}(3c + \beta)\alpha_k ,$$

thanks to Lemma 12. If $k$ is the index of a successful iteration, we can come back to the last unsuccessful iteration $k'$, since

$$\|\nabla f(x_k)\| \leq \frac{\beta}{a} \|\nabla f(x_{k'})\| \leq \frac{\beta}{a} \frac{1}{2\kappa}(3c + \beta)\alpha_{k'} \leq \frac{\beta}{a} \frac{1}{2\kappa}(3c + \beta)\left(\frac{\alpha_k}{\theta}\right) ,$$

where the first inequality comes from Lemma 11, the second from Lemma 12 and the third from the fact that $\alpha_k \geq \theta \alpha_{k'}$. $\qquad\square$

### C.3 Regret Analysis of FDS-Plan

**Lemma 16.** *If $f$ satisfies the assumptions of Theorem 4 and on $\cap_{k \leq K} \mathcal{E}_k$,*

$$\tilde{R}_K \leq C_4 log(2/\delta) + C_5 \log(2/\delta) \left(\sum_{k=1}^{K} N_k\right)^{2/3} ,$$

*where* $\begin{cases} C_4 = \frac{32}{c^2} C_1 \alpha_0^{-4}(S_{\mathbb{D}} + 1) \\ C_5 = \frac{32}{c^2}(S_{\mathbb{D}} + 1)\left(\frac{c}{32} \frac{1}{(1-\theta^2)}(f(x_1) - f(x_\star) + \rho(\alpha_0))\right)^{1/3}\left(\frac{1}{a}\eta_2^2 + \eta_2 + \beta\right) .\end{cases}$

*Proof.* In the following we study the case where $\cap_{k \leq K} \mathcal{E}_k$ holds true.

As in the deterministic case, we decompose the regret as

$$\tilde{R}_K = \sum_{k=0}^{k_f} N_k \left( f(x_k) - f(x_\star) + \sum_{v \in \mathbb{D}} f(x_k + \alpha_k v) - f(x_\star) \right)$$
$$+ \sum_{k=k_f}^{K} N_k \left( f(x_k) - f(x_\star) + \sum_{v \in \mathbb{D}} f(x_k + \alpha_k v) - f(x_\star) \right) .$$

We start by dealing with the cumulative regret before $k_f$. We write

$$\sum_{k=0}^{k_f} N_k \left( f(x_k) - f(x_\star) + \sum_{v \in \mathbb{D}} f(x_k + \alpha_k v) - f(x_\star) \right)$$

$$\leq N_0 \sum_{k=0}^{k_f} \left( f(x_k) - f(x_\star) + \sum_{v \in \mathbb{D}} f(x_k + \alpha_k v) - f(x_\star) \right)$$

$$\leq \frac{32}{c^2} \alpha_0^{-4} \log(2/\delta) \sum_{k=0}^{k_f} \left( f(x_k) - f(x_\star) + \sum_{v \in \mathbb{D}} f(x_k + \alpha_k v) - f(x_\star) \right)$$

$$\leq \frac{32}{c^2} \alpha_0^{-4} \log(2/\delta) \times 2C_1$$

$$= C_4 \log(2/\delta),$$

where the last inequality is obtained exactly as in the proof of Lemma 5 with the help of Lemma 14 instead of Lemma 4. By using the above inequality and the decomposition of the regret, we get

$$\tilde{R}_K - C_4 \log(2/\delta)$$

$$\leq \sum_{k=k_f}^{K} N_k \left( f(x_k) - f(x_\star) + \sum_{v \in \mathbb{D}} f(x_k + \alpha_k v) - f(x_\star) \right)$$

$$\leq \sum_{k=k_f}^{K} N_k \left( f(x_k) - f(x_\star) + \sum_{v \in \mathbb{D}} f(X_k + \alpha_k v) - f(x_k) + f(x_k) - f(x_\star) \right)$$

$$\leq \sum_{k=k_f}^{K} N_k \left( (S_{\mathbb{D}} + 1)(f(x_k) - f(x_\star)) + \sum_{v \in \mathbb{D}} f(x_k + \alpha_k v) - f(x_k) \right)$$

$$\leq (S_{\mathbb{D}} + 1) \sum_{k=k_f}^{K} N_k \left( \frac{1}{a} \|\nabla f(x_k)\|^2 + \|\nabla f(x_k)\| \alpha_k + \beta \alpha_k^2 \right),$$

where $C_4 = \frac{32}{c^2} C_1 \alpha_0^{-4} (S_{\mathbb{D}} + 1)$. The fourth inequality comes from the regularity assumptions required for Theorem 2 together with Lemma 10. We use Lemma 15 to get that for any $k > k_f$,

$$\|\nabla f(x_k)\| \leq \eta_2 \alpha_k.$$

Then

$$\frac{1}{a} \|\nabla f(x_k)\|^2 + \|\nabla f(x_k)\| \alpha_k + \beta \alpha_k^2 \leq \frac{1}{a} \eta_2^2 \alpha_k^2 + \eta_2 \alpha_k^2 + \beta \alpha_k^2.$$

We get

$$\sum_{k=k_f}^{K} N_k \left( \frac{1}{a} \|\nabla f(x_k)\|^2 + \|\nabla f(x_k)\| \alpha_k \right) \leq C_6 \log(2/\delta) \sum_{k=k_f}^{K} (\alpha_k)^{-2}$$

$$\leq C_6 \log(2/\delta) \sum_{k=0}^{K} (\alpha_k)^{-2}.$$

where $C_6 = \left( \frac{1}{a} \eta_2^2 + \eta_2 + \beta \right) \frac{32}{c^2}$. Consequently

$$\tilde{R}_K \leq C_4 \log(2/\delta) + C_6 \log(2/\delta) \sum_{k=0}^{K} (\alpha_k)^{-2}.$$

The number of function evaluations is defined as

$$\sum_{k=0}^{K} N_k = \frac{32 \log(2/\delta)}{c} \sum_{k=0}^{K} \alpha_k^{-4}.$$

Thanks to Lemma 13,

$$\sum_{k=0}^{K}(\alpha_k)^2 \leq \frac{2}{c(1-\theta^2)}\left(f(x_1) - f(x_\star) + \rho(\alpha_0)\right).$$

By Hölder's inequality, we get

$$\sum_{k=0}^{K}(\alpha_k)^{-2} = \sum_{k=0}^{K}(\alpha_k)^{2/3}(\alpha_k)^{-2/3-2}$$

$$\leq \left(\sum_{k=0}^{K}(\alpha_k)^{2/3\times3}\right)^{1/3}\left(\sum_{k=0}^{K}(\alpha_k)^{-8/3\times3/2}\right)^{2/3}$$

$$\leq \left(\sum_{k=0}^{K}(\alpha_k)^2\right)^{1/3}\left(\sum_{k=0}^{K}(\alpha_k)^{-4}\right)^{2/3}.$$

And thus

$$\tilde{R}_K - C_4\log(2/\delta)$$

$$\leq C_6\log(2/\delta)\left(\frac{2}{c(1-\theta^2)}\left(f(x_1) - f(x_\star) + \rho(\alpha_0)\right)\right)^{1/3}\left(\frac{c}{8\log(2/\delta)}\sum_{k=1}^{K}N_k\right)^{2/3}$$

$$\leq C_5\log(2/\delta)\left(\frac{1}{\log(2/\delta)}\sum_{k=1}^{K}N_k\right)^{2/3}$$

$$\leq C_5\log(2/\delta)^{1/3}\left(\sum_{k=1}^{K}N_k\right)^{2/3},$$

where $C_5 = (S_{\mathbb{D}}+1)C_6\left(\frac{c}{32}\frac{1}{(1-\theta^2)}(f(x_1) - f(x_\star) + \rho(\alpha_0))\right)^{1/3}$.

$\square$

*Proof.* **of Theorem 4** We note $K_T$ the last round reached by the algorithm with $T$ evaluations. Lemma 16 proves that on $\cap_{k\leq K_T}\mathcal{E}_k$,

$$\tilde{R}_{K_T} \leq C_4\log(2/\delta) + C_5\left(\frac{1}{(S_{\mathbb{D}}+1)}\right)^{2/3}\log(2/\delta)^{2/3}(T)^{2/3}.$$

Thanks to Lemma 8,

$$\mathbb{P}\left(\cup_{k=1}^{K_T}\mathcal{E}_k^C\right) \leq \sum_{k=1}^{T}\mathbb{P}\left(\mathcal{E}_k^C\right) \leq (S_{\mathbb{D}}+1)\sum_{t=1}^{T}T^{-4/3}/2 \leq (S_{\mathbb{D}}+1)T^{-1/3}, \tag{5}$$

when taking $\delta = T^{-4/3}$, since $K_T \leq T$. Hence,

$$\mathbb{E}[R_T] \leq \frac{4}{3}C_4\log(2T) + \frac{4}{3}\left(\frac{1}{(S_{\mathbb{D}}+1)}\right)^{2/3}C_5\log(2T)^{2/3}T^{2/3} + (S_{\mathbb{D}}+1)UT^{2/3}$$

$$= O((\log T)^{2/3}T^{2/3}).$$

$\square$

## C.4 Regret Analysis of FDS-Seq

Instead of considering $\mathcal{E}_k$ as in the previous section, we need to consider $\mathcal{E}'_k = \{(f(x_k) - f(x_k + \alpha_k v_k) \leq 3c/2(\alpha_k)^2$ and $k$ is an unsuccessful iteration) or ($k$ is a successful iteration and $f(x_k) - f(x_k + \alpha_k v_k) \geq c/2(\alpha_k)^2)\}$. Instead of Lemma 8 we prove the following result.

**Lemma 17.** *The probability of $\mathcal{E}'_k$ is lower bounded by*

$$\mathbb{P}\left(\mathcal{E}'_k | \mathcal{F}_{k-1}\right) \geq 1 - \delta \times N_k^2 \times (|\mathbb{D}|+1) \,,$$

*where $\mathcal{F}_{k-1}$ is the $\sigma$-field representing the history.*

*Proof.* Fix $v \in \mathbb{D}$. First assume that $f(x_k) > f(x_k + \alpha_k v) + 3/2\rho(\alpha_k)$. In particular, $f(x_k) > f(x_k + \alpha_k v) + \rho(\alpha_k)$. We denote $n_{0,k}^\tau$ and $n_{v,k}^\tau$ the values of $n_{0,k}$ and $n_{v,k}$ at the end of the while loop of FDS-Seq. Observe that

$$\mathbb{P}\left(\mathcal{E}'_k | \mathcal{F}_{k-1}, n_{0,k}^\tau = n_{v,k}^\tau = N_k\right)$$
$$\geq 1 - \delta \times (|\mathbb{D}|+1) \,.$$

Hence we only need to focus on the case when the first part of Condition 1 is first satisfied. In this case, knowing $n_{0,k}^\tau, n_{v,k}^\tau$, the probability that $\hat{f}_{n_{0,k}^\tau}(x_k) < \hat{f}_{n_{v,k}^\tau}(x_k + \alpha_k v) + \rho(\alpha_k)$ when the first row of Condition 1 is first satisfied is bounded as follows. We have

$$\mathbb{P}\left( f(x_k) - \hat{f}_{n_{0,k}^\tau}(x_k) - (f(x_k + \alpha_k v) - \hat{f}_{n_{v,k}^\tau}(x_k + \alpha_k v)) \geq \right.$$
$$\left. \sqrt{2\sigma^2 \log(1/\delta)}\sqrt{\frac{1}{n_{0,k}^\tau} + \frac{1}{n_{v,k}^\tau}} \middle| \mathcal{F}_k, n_{0,k}^\tau, n_{v,k}^\tau \right) \leq \delta.$$

Since $f(x_k) > f(x_k + \alpha_k v) + \rho(\alpha_k)$,

$$f(x_k) - \hat{f}_{n_{0,k}^\tau}(x_k) - (f(x_k + \alpha_k v) - \hat{f}_{n_{v,k}^\tau}(x_k + \alpha_k v))$$
$$\geq -\rho(\alpha_k) - \hat{f}_{n_{0,k}^\tau}(x_k) + \hat{f}_{n_{v,k}^\tau}(x_k + \alpha_k v) \,.$$

So that the above deviation bound results in:

$$\mathbb{P}\left( \hat{f}_{n_{0,k}^\tau}(x_k) - \hat{f}_{n_{v,k}^\tau}(x_k + \alpha_k v) - \rho(\alpha_k) \leq \right.$$
$$\left. -\sqrt{2\sigma^2 \log(1/\delta)}\sqrt{\frac{1}{n_{0,k}^\tau} + \frac{1}{n_{v,k}^\tau}} \middle| \mathcal{F}_k, n_{0,k}^\tau, n_{v,k}^\tau \right)$$

Finally, we apply a union bound. Since $n_{0,k}^\tau$ and $n_{v,k}^\tau$ both belong to $[0, N_k]$ and cannot be simultaneously equal to $N_k$:

$$\mathbb{P}\left( \hat{f}_{n_{0,k}^\tau}(x_k) - \hat{f}_{n_{v,k}^\tau}(x_k + \alpha_k v) - \rho(\alpha_k) \leq \right.$$
$$\left. -\sqrt{2\sigma^2 \log(1/\delta)}\sqrt{\frac{1}{n_{0,k}^\tau} + \frac{1}{n_{v,k}^\tau}} \middle| \mathcal{F}_k, \ \text{not}(n_{0,k}^\tau = n_{v,k}^\tau = N_k) \right) \leq (N_k^2 - 1)\delta \,.$$

This amounts to a bound of the probability of $k$ being an unsuccessful iteration and thus of $\mathcal{E}'^C_k$, when the first part of condition 1 is satisfied. Summing with the probability of $\mathcal{E}'^C_k$ in the other case, we obtain $\mathbb{P}\left(\mathcal{E}'(k)\right) \geq N_k^2 \delta$.

The case $f(x_k) > f(x_k + \alpha_k v) + 3/2\rho(\alpha_k)$ can be treated in the exact same way.

$\square$

To adapt the proof of Theorem 4 to FDS-Seq (with a different choice of $\delta = T^{-10/3}$), it suffices to replace Equation 5 by

$$\mathbb{P}\left(\cup_{k=1}^{K_T} \mathcal{E'}_k^C\right) \leq (S_{\mathbb{D}} + 1) \sum_{k=1}^{T} \mathbb{P}\left(\mathcal{E'}_k^C\right) \leq (S_{\mathbb{D}} + 1) \sum_{k=1}^{K_T} N_k^2 T^{-10/3}$$

$$\leq (S_{\mathbb{D}} + 1) \sum_{k=1}^{T} T^2 T^{-10/3}/2 \leq (S_{\mathbb{D}} + 1) T^{-1/3} \ .$$

The regret is hence

$$\mathbb{E}[R_T] \leq \frac{10}{3} C_4 \log(2T) + \frac{10}{3}\left(\frac{1}{(S_{\mathbb{D}} + 1)}\right)^{2/3} C_5 \log(2T)^{2/3} T^{2/3} + (S_{\mathbb{D}} + 1)UT^{2/3}$$

$$= O((\log T)^{2/3} T^{2/3}) \ .$$

## D  Noisy and Constrained Set-Up

In this section we analyze the behavior of the algorithms in the presence of linear constraints. The complexity of feasible direct search with linear constraints in the noiseless case has been studied by Gratton et al. (2019). In this paper, instead of studying the speed at which the gradient converges to 0 as is usual in the unconstrained case, the authors study the convergence of a lower bound of the gradient $\chi(x) := \max_{x+v \in \mathcal{D}, \|v\| \leq 1} -\nabla f(x)^T v$ to 0. Indeed, the convergence of the gradient to 0 might be unachievable when the optimum lies on the boundaries, but $\chi(x)$ is equal to 0 if and only if $x$ is optimal. The paper proves that the first iteration $K_\epsilon$ at which $\chi(x_k)$ is smaller than $\epsilon$ is of the order of $\epsilon^{-2}$, like in the unconstrained case.

In the following, we denote by $U$ the global upper bound of $f(x) - f(x_\star)$ on the domain.

### D.1  Intermediate Results

We recall that $\mathcal{E}_k$ denotes the event

$$\mathcal{E}_k = \{|f(x_k + \alpha_k v) - \hat{f}(x_k + \alpha_k v)| \leq c/4(\alpha_k)^2\}, \ \forall v \in \mathbb{D} \cup \{0\}\}.$$

Lemmas 8, 9, 10 are left unchanged by the transition to constrained domains. A version of Lemma 3.4. of (Gratton et al., 2019) reads :

**Lemma 18.** *On $\cap_{k \leq K} \mathcal{E}_k$, and if $f$ satisfies Assumption 1, then the following holds: if the k-th iteration is unsuccessful, then*

$$\chi(x_k) \leq \left(\frac{\beta}{2\kappa} + \frac{Bg}{\eta_{\min}}\right)\alpha_k + \frac{3\rho(\alpha_k)}{2\kappa\alpha_k} := L_2\alpha_k,$$

*where $\eta_{\min} := \lambda(N)$ where $N$ is the set of all possible approximate normal cones $N(x, \alpha)$, $\forall \in \mathcal{D}$, $\alpha \in \mathbb{R}^d$.*

*Proof.* It is straightforward to prove

$$\|P_{T(x_k,\alpha_k)}(-\nabla f(x_k))\| \leq \frac{1}{\kappa}\left(\frac{\beta}{2}\alpha_k + \frac{3\rho(\alpha_k)}{2\alpha_k}\right) = \frac{1}{2\kappa}\left(\beta\alpha_k + 3c\alpha_k^2\right),$$

with the same elements as in the proof Lemma 11, by noticing that $\mathbb{D}_k$ contains $\mathcal{G}_k$, that generates $T(x_k, \alpha_k)$. To prove the bound on $\chi(x_k)$, we use the Moreau decomposition, stating that any vector $v \in \mathbb{R}^d$ can be

decomposed as $v = P_{T_k}[v] + P_{N_k}[v]$ with $N_k = N(x_k, \alpha_k)$ and $P_{T_k}[v] > P_{N_k}[v] = 0$, and write

$$\chi(x_k) = \max_{x+v \in \mathcal{D}, \, \|v\| \leq 1} (P_{T_k}[-\nabla f(x_k)] + (P_{T_k}[v]v^T + P_{N_k}[v])^T P_{N_k}[-\nabla f(x_k)])$$

$$\leq \max_{x+v \in \mathcal{D}, \, \|v\| \leq 1} (v^T P_{T_k}[-\nabla f(x_k)] + P_{N_k}[v]^T P_{N_k}[-\nabla f(x_k)])$$

$$\leq \max_{x+v \in \mathcal{D}, \, \|v\| \leq 1} \|P_{T_k}[-\nabla f(x_k)]\| + \|P_{N_k}[v]\| \|P_{N_k}[-\nabla f(x_k)]\|. \tag{6}$$

The first term of the right hand side of Equation 6 is bounded in the following way

$$\|P_{T(x_k, \alpha_k)}(-\nabla f(x_k))\| \leq \frac{1}{2\kappa} \left( \beta \alpha_k + 3c\alpha_k^2 \right)$$

consequently.

**Lemma 19** (Proposition B.1 of (Lewis & Torczon, 2000)). *Let $x \in \mathcal{D}$ and $\alpha > 0$. Then, for any vector $v$ such that $x + v \in \mathcal{D}$, one has*

$$\|P_{N(x,\alpha)}[v]\| \leq \frac{\alpha}{\eta_{\min}}.$$

This in turn provides a bound of the second term of the right hand side of Equation 6:

$$\|P_{N_k}[v]\| \|P_{N_k}[-\nabla f(x_k)]\| \leq \frac{\alpha}{\eta_{\min}} B_g,$$

which suffices to conclude the proof. $\qquad \square$

**Lemma 20.** *Assume that $\cap_{k \leq K} \mathcal{E}_k$ holds, and $f$ satisfies Assumption 1. Set $\epsilon > 0$. Let $h$ denote the mapping from $\epsilon$ to*

$$h(\epsilon) = \left( \frac{2L_2^2 U}{c\theta^1} \right) \epsilon^{-2} + \frac{\log(\frac{\alpha_0 L_2}{\theta})}{\log(1/\theta)} + \frac{2U}{\alpha_0^2} = E_1 \epsilon^{-2} + E_2,$$

*where $E_1 = \left( \frac{2L_2^2 U}{c\theta^1} \right)$ and $E_2 = \frac{\log(\frac{\alpha_0 L_2}{\theta})}{\log(1/\theta)} + \frac{2U}{\alpha_0^2}$. Denote by $k(\epsilon)$ the first iteration of the algorithm where $\chi(x_k) \leq \epsilon$. If $h(\epsilon) \leq K$, then*

$$k(\epsilon) \leq h(\epsilon).$$

The proof is a mere adaptation of the proof of Theorem 1 of (Gratton et al., 2015), with different constants (we use Lemma 18 and 13).

**Lemma 21.** *If $f$ satisfies Assumption 1,*

$$a \|x_k - x_\star\| \leq \chi(x_k).$$

*Proof.*

$$a \|x_k - x_\star\| \leq \frac{f(x) - f(x_\star)}{\|x_k - x_\star\|} \leq \frac{-\nabla f(x)(x_\star - x_k)}{\|x_k - x_\star\|} \leq \chi(x_k),$$

by definition of $\chi(x_k)$. $\qquad \square$

**Lemma 22.** *If $f$ satisfies Assumption 1 and if $x_\star$ is in the interior of $\mathcal{D}$ and the algorithm achieves descent at each iteration, then*

$$\forall k' > k, \|\chi(x_{k'})\| \leq \left( \frac{\beta}{a} \right)^{3/2} \|\chi(x_k)\|.$$

*Proof.* Like in the proof of Lemma 1, we get

$$a \|x_k - x^*\| \leq \|\nabla f(x_k)\|.$$

Hence,

$$\beta\|x_{k'} - x_\star\| \leq \frac{\beta}{\sqrt{a}}\sqrt{f(x_{k'}) - f(x_\star)} \leq \frac{\beta}{\sqrt{a}}\sqrt{f(x_k) - f(x_\star)}$$

$$\leq \frac{\beta}{\sqrt{a}}\sqrt{\beta(x_k - x_\star)^2} \leq \frac{\beta^{3/2}}{\sqrt{a}}\sqrt{\frac{\chi(x_k)^2}{a}}$$

$$\leq \left(\frac{\beta}{a}\right)^{3/2}\chi(x_k),$$

where the first inequality comes from the strong convexity, and the second one comes from the fact that the algorithm is a descent, the third one comes from Lemma 21.

Now let $v = \arg\max_{x+v\in\mathcal{D},\ \|v\|\leq 1} -v^T\nabla f(x_{k'})$.

$$-v^\top\nabla f(x_{k'}) = -v^\top\nabla f(x_{k'}) + v^\top\nabla f(x_*) \leq \|\nabla f(x_{k'}) - \nabla f(x_*)\| \leq \beta\|x_k - x_\star\|,$$

because $\nabla f(x_*) = 0$. This concludes the proof. □

## D.2 Regret Analysis when the Optimum is in the interior of $\mathcal{D}$

Let us assume that $x_\star$ is in the interior of $\mathcal{D}$. Let us denote by $\Delta$ the distance from $x_\star$ to the closest boundary, and by $r = \Delta/4$.

**Lemma 23.** *If $\chi(x) \leq ar$ then $\|x - x_\star\| \leq r$.*

If $\|x - x_\star\| \geq r$ then $r \leq \frac{1}{a}\chi(x)$ thanks to Lemma 21.

**Lemma 24.** *For any $k \geq k\left((a/\beta)^{3/2}ar\right)$, $\|x_k - x_\star\| \leq r$.*

Thanks to Lemma 1, after $k\left((a/\beta)^{3/2}ar\right)$ iterations, $\chi(x_k) \leq ar$. And thanks to the previous lemma, we thus have $\|x_k - x_\star\| \leq r$.

**Lemma 25.** *Set $k_s$ the index of the first successful iteration following $k\left((a/\beta)^{3/2}ar\right)$ where $\alpha_k \leq \Delta/2$. After iteration $k_s$, $T(x_k, \alpha_k)$ spans all directions in $\mathbb{R}^d$, so that the instantaneous regret is the same as that of the algorithm in the unconstrained case with initial point $x_{k_s}$ and initial step-size $\alpha_{k_s}$. The iteration of this first successful iteration comes before $k_i := k\left((a/\beta)^{3/2}ar\right) + \frac{\log(\alpha_0/\Delta)}{\log 1/\theta}$.*

*Proof.* If $k$ is a successful iteration $\alpha_k \leq 2r = \Delta/(2)$, since $\|x_k - x_\star\| \leq r$ and $\|x_{k+1} - x_\star\| \leq r$. And if $k$ is an unsuccessful iteration, it comes after one of those successes and a sequence of unsuccessful iterations, which yields $\alpha_k \leq \Delta/(2)$. □

**Lemma 26.** *On $\cap_{k\leq K}\mathcal{E}_k$,*

$$\tilde{R}_K \leq C_7\log(2/\delta)(1/\theta)^{-4C_f} + C_5\log(2/\delta)^{1/3}\left(\sum_{k=1}^K N_k\right)^{2/3},$$

*where $C_7 = (S_\mathbb{D} + 1)U\frac{32}{c^2\alpha_0^4}\frac{1}{(1/\theta)-1}$ and*

$$C_f = E_1\left(\left(\frac{a}{\beta}\right)^{3/2}ar\right)^{-2} + E_2 + \frac{\log(\alpha_0/\Delta)}{\log 1/\theta} + \frac{\beta\Delta^2}{\alpha_0}.$$

*Proof.* In the proof of Lemma 16, we isolated the steps preceding the first unsuccessful iteration. Similarly here, we treat the iterations before the first unsuccessful iteration after $k_i$, denoted by $k'_f$, separately from other iterations.

$$\tilde{R}_K \leq \sum_{k=0}^{k'_f} N_k(f(x_k) - f(x_\star)) + \sum_{k=0}^{k'_f} \left( N_k \sum_{v \in \mathbb{D}_k} f(x_k + \alpha_k v) - f(x_\star) \right)$$
$$+ \sum_{k=k'_f}^{K} (N_k(|\mathbb{D}_k|+1)(f(x_k) - f(x_\star))) + \sum_{k=k'_f}^{K} N_k \left( \sum_{v \in \mathbb{D}_k} f(x_k + \alpha_k v) - f(x_k) \right).$$

Because $f(x) - f(x_\star)$ is bounded by $U$,

$$\sum_{k=0}^{k'_f} N_k(f(x_k) - f(x_\star)) + \sum_{k=0}^{k'_f} \left( N_k \sum_{v \in \mathbb{D}_k} f(x_k + \alpha_k v) - f(x_\star) \right)$$
$$\leq \sum_{k=0}^{k'_f} (|\mathbb{D}_k|+1) N_k U.$$

By rewriting $N_k$,

$$\sum_{k=0}^{k'_f} (|\mathbb{D}_k|+1) N_k U \leq \sum_{k=0}^{k'_f} (|\mathbb{D}_k|+1) U \frac{32 \log(1/\delta)}{c^2 \alpha_k^4}$$
$$\leq \sum_{k=0}^{k'_f} (|\mathbb{D}_k|+1) U \frac{32 \log(1/\delta)}{c^2 \alpha_0^4 \theta^{4k}}$$
$$\leq (S_\mathbb{D} + 1) U \frac{32 \log(1/\delta)}{c^2 \alpha_0^4} \frac{(1/\theta)^{-4k'_f}}{(1/\theta) - 1}.$$

Also on $\cap_{k \leq K} \mathcal{E}_k$,

$$k'_f \leq k_i + \frac{f(x_{k_i}) - f(x_\star)}{\alpha_0} \leq k_i + \frac{\beta \Delta^2}{\alpha_0} \leq k \left( \left( \frac{a}{\beta} \right)^{3/2} ar \right) + \frac{\log(\alpha_0/\Delta)}{\log 1/\theta} + \frac{\beta \Delta^2}{\alpha_k},$$

where the first inequality comes from the same argument used to prove Lemma 4, the second inequality comes from the smoothness of $f$ and the third one comes from the definition of $k_i$.

On $\cap_{k \leq K} \mathcal{E}_k$,

$$k \left( \left( \frac{a}{\beta} \right)^{3/2} ar \right) \leq E_1 \left( \left( \frac{a}{\beta} \right)^{3/2} ar \right)^{-2} + E_2,$$

with $E_1$, $E_2$ defined in (Gratton et al., 2019). Finally, we focus on the part of the regret accumulated before $k'_f$. On $\cap_{k \leq K} \mathcal{E}_k$

$$\sum_{k=k'_f}^{K} (N_k(|\mathbb{D}_k|+1)(f(x_k) - f(x_\star))) + \sum_{k=k'_f}^{K} N_k \left( \sum_{v \in \mathbb{D}_k} f(x_k + \alpha_k v) - f(x_k) \right)$$
$$\leq C_5 \log(2/\delta)^{1/3} \left( \sum_{k=1}^{K} N_k \right)^{2/3},$$

by following exactly the same steps as those needed to bound the regret in the unconstrained case. $\qquad\square$

**Theorem 2.** *Under Assumptions 1, 3 and 4, and if $x_\star \in int(\mathcal{D})$, the cumulative regret $R_T$ of FDS-Plan (respectively FDS-Seq) after the first $T$ evaluations of $f$, satisfies*

$$\mathbb{E}[R_T] = O(\log(T)^{2/3}T^{2/3})$$

*for the choice $\delta = T^{-4/3}$ (respectively $\delta = T^{-10/3}$ for FDS-Seq).*

*Proof.* **for FDS-Plan.** We denote by $K_T$ the last round reached by the algorithm with $T$ evaluations.. Lemma 26 proves that on the event $\cap_{k \leq K_T} \mathcal{E}_k$,

$$\tilde{R}_K \leq C_7 \log(2/\delta)(1/\theta)^{-4C_f} + C_5 \log(2/\delta)^{1/3} \left( \sum_{k=1}^{K} N_k \right)^{2/3}.$$

Thanks to Lemma 8,

$$\mathbb{P}\left( \cup_{k=1}^{K_T} \mathcal{E}_k^C \right) \leq (S_{\mathbb{D}} + 1)\mathbb{P}\left( \cup_{k=1}^{T} \mathcal{E}_k^C \right) \leq (|\mathbb{D}_k|+1) \sum_{t=1}^{T} T^{-4/3}/2 \leq (S_{\mathbb{D}} + 1)T^{-1/3}$$

since $K_T \leq T$. Hence,

$$\begin{aligned}
\mathbb{E}[R_T] &\leq \frac{4}{3}C_7(1/\theta)^{-4C_f}\log(2T) + \frac{4}{3}\left( \frac{1}{(|\mathbb{D}_k|+1)} \right)^{2/3} C_5 \log(2T)^{2/3}T^{2/3} \\
&\quad + ((S_{\mathbb{D}} + 1))UT^{2/3} \\
&= O((\log T)^{2/3}T^{2/3}).
\end{aligned}$$

$\square$

**Adaptation of the proof for FDS-Seq** The way of adapting the proof of FDS-Plan to the case of FDS-Seq of Section C.4 applies verbatim.

# E  Details on the implementation of HOO in Section 2.2

To implement HOO in the simulations of Section 2.2, the tree of partitions that we used is built in the following way. We set the parameter $\rho$ of HOO as suggested by Bubeck et al. (2011) to $2^{-2/d}$. A binary tree of depth $H = \frac{\log(1/T)}{2\log(\rho)}$ of partitions of $[0,1]^d$ is obtained by recursively halving the cells at each depth $h$ of the tree along dimension $h \pmod 2$. At depth $h$, this approach yields a partition formed by rectangular cells represented by their lower left corner $[a_i, b_i]$. Then, in order to remove unwanted cells, we traverse the tree, starting from the leaves, and remove every cell having an empty intersection with the domain. Due to the geometry of the simplex, knowing if a cell intersects the domain boils down to checking if its representation $[a_i, b_i]$ belongs to it. When the algorithm selects cell $(h, i)$ at time $t$, the representation of that cell is chosen as a sampling point. In the simulation, the smoothness parameter $\nu_1$ of HOO is set to 16.

# F  Additional Experiments

Here, as in Section 2.2, we focus on the case in which there are three resources $(d = 2)$. The loss functions for resources 1 and 3 are of the same form as in Section 2.2 and $w_i(x) = -\tau_i \frac{\log(1+\gamma x)}{\log(1+\gamma)}$ with $\gamma = 2$, $\tau_1 = 1$, $\tau_3 = 0.3$, but now the second resource is associated to $w_2(x) = 0.1\,x$. This choice of reward functions results in an optimal choice whose second component is zero. We set the horizon to $T = 100,000$ and use a Gaussian noise with standard deviation $\sigma = 0.1$.

We show the trajectories of FDS-Plan and FDS-Seq in Figure 3. Notice that the trajectories do not change drastically compared to those of Section 2.2, which seems to indicate that the location of the optimal

allocation on the border of the feasible set is not a problem in practice. We complement these plots with regret plots (Figure 4) of all the algorithms detailed in Section 2.2 run first on the environment described in this same section and second on the environment described above with the optimum on the border of the simplex. Once again, this seems to show that the optimum lying in the border is not an issue in practice. Incidentally, this last plot also shows that, as expected, UCB should be the preferred algorithm in dimension $d = 2$, as it is simpler and gives excellent results.

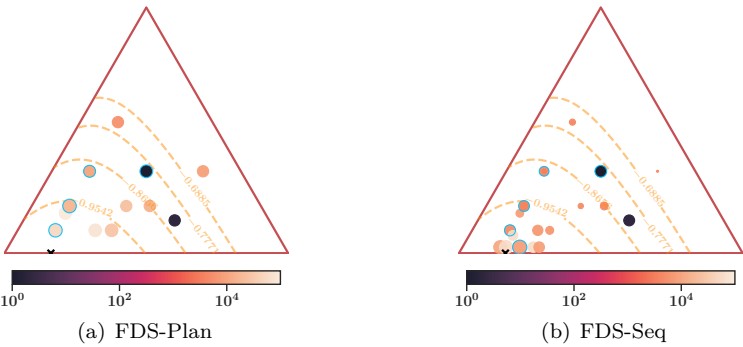

(a) FDS-Plan           (b) FDS-Seq

Figure 3: Single trajectories with the optimum on the border

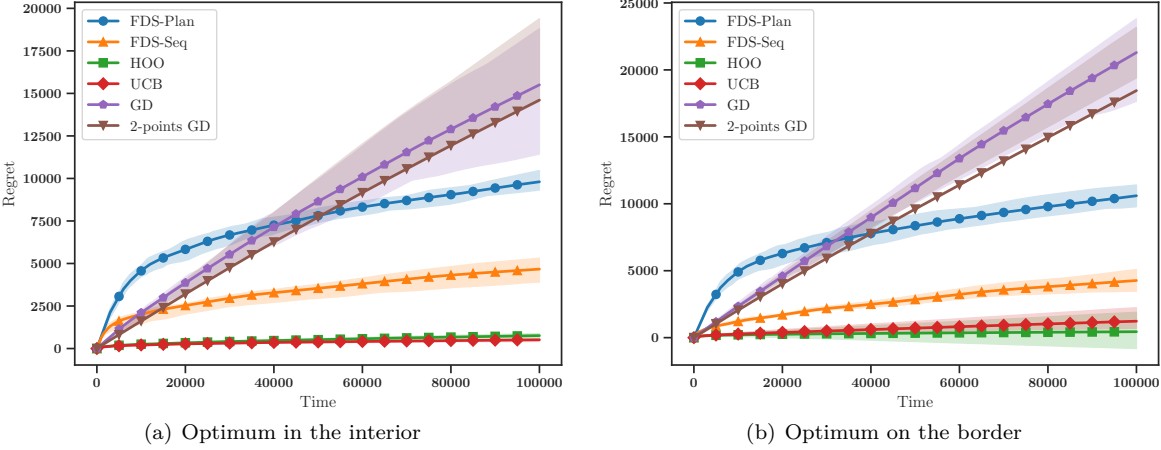

(a) Optimum in the interior           (b) Optimum on the border

Figure 4: Regret plots in dimension $d = 2$

