# OpenReview forum: "Stochastic Direct Search Methods for Blind Resource Allocation"
_TMLR — Accepted by TMLR_

### Review · Reviewer_CKgM · 2024-01-19

**Summary Of Contributions:**

The paper considers the problem of online bandits/zeroth order optimization under resource constraints. It develops three natural algorithms and derives the regret bound accordingly.

**Audience:**

Yes

**Broader Impact Concerns:**

NaN.

**Claims And Evidence:**

Yes

**Requested Changes:**

See above for questions.

**Strengths And Weaknesses:**

The formulation considered in the paper seems to be new and different from the existing ones in the literature. Yet I have the following concerns:

1. The formulation seems niche in that there is a rich literature for both online bandit convex optimization and online resource allocation. The paper considers a formulation combining the two settings. I am not quite clearly motivated as to why this formulation matters:

- Resource constraints are imposed individually for each time step. This is equivalent to restricting each $x_t$ to belong to some set $\mathcal{S}$ for all the $t$. First, I don't know how much this changes the nature of the online bandit convex optimization problem, and more specifically, why the existing online bandit convex optimization algorithms cannot be applied here. Second, the more common resource constraints seem to be $x_1+....+x_T\le B$, and this type of cumulative constraint will significantly differentiate from the online bandits optimization literature. To me, the cumulative budget constraint more naturally reflects the ads problem where the budget is the bidding budget for each advertiser.

- There is a lack of justification for why the cost function $f$ has this additive form.

2. The technical contribution. The analysis seems very standard to me. I wonder what are the technical challenges of the analysis. And as mentioned in the paper, "in the absence of a lower bound, the optimality of such a regret rate is unsure." It is hard to appreciate the optimality of the algorithm.

3. The scalability with the dimension $d$. Assumption 2 may result in an undesirable dependence for the PSS set $D$ on the dimension $d$. Authors should consider to derive some bound on the relationship between the cardinality of $D$, the parameter $\kappa$, and the dimension $d$. For the numerical experiments, it is more of within a low-dimensional regime.

Minor comments:

- Seems no definition for the abbreviation PSS.
- $\theta<1$ instead of $\theta\le1$  in Algorithm 1.

---

> ### Author Response · Authors · 2024-03-05
> **Response to review**
>
> We thank the reviewer for their comments on the manuscript and answer to specific points below.
>
> 1. About step-level constraints: Step-level constraints are crucial in scenarios requiring a steady pace, such as programmatic advertising where advertisers must balance their marketing spendings over a specified period, usually by setting a daily budget. Considering that a round corresponds to a day, our setting models the optimization problem faced when distributing the daily budget into sub-campaigns. In this application, the additive form of the objective function corresponds to the natural summation over the returns of all sub-campaigns. Cumulative and step-level constraints lead to distinct optimization problems, none of which being a reduction of the other.
> 2. About the significance of the contribution: As mentioned in the manuscript, we are not aware of any prior work that  presents an analysis of direct search algorithms in terms of regret. Naturally, the analysis that we propose combines arguments found both in the literature on direct search algorithms and in works on continuously-armed stochastic bandits. A significant technical challenge  is that, while in traditional analyses of direct-search, the number of rounds is fixed and the analysis proceeds by looking at the distance to the optimum at each round, here, the number of rounds is random (the indexing of the regret is the actual number of function evaluations not the number of rounds).
> 3. Scalability in dimension $d$: The computational aspect of the question (including the dependence of the cardinality of the PSS on d) is discussed at the end of the first section 3.2. From the definition of $\kappa$ it is clear that it tends to decrease as $\sqrt{d}$. This being said, this is of course not the only dependence in d. We agree with the reviewer that zeroth-order’s methods are likely to become inefficient in very high dimensions (i.e., more than 50). This being said, dimensions of the order of, say,  5 are common in programmatic advertising and correspond roughly to the illustrations shown in the paper (although we focus on the case where d=2 mainly for illustration purposes).

---

### Review · Reviewer_7z8h · 2024-01-30

**Summary Of Contributions:**

This paper studies a problem called blind resource allocation problem, which is an instance of convex optimization with linear constraints under bandit feedback and with noisy function evaluations. Specifically, this a sequential decision-making problem where the learner receives the function value corresponding to the queried point but corrupted by a zero-mean sub-gaussian noise (under the conditional independence assumption commonly considered in the literature), where for concreteness the noise is taken to by Gaussian. The learner has prior knowledge on the variance of the noise, but no access to the gradient information. The performance measure is a notion of regret that is defined in terms of function evaluations, unlike those of existing literature which is defined in terms of decision rounds. The paper presents two algorithms for this problem, FDS-Plan and FDS-Seq. These algorithms are variants of the direct search method, which are derivative-free methods from the optimization literature for similar problems. The first algorithm relies on pre-computed numbers of evaluations whereas the second decides these numbers adaptively. Both algorithms are analyzed under the assumption of strong-convexity and smoothness of the underlying objective function, and are shown to enjoy a regret growing as $T^{2/3}$ (in order, and ignoring logarithmic factors) after $T$ many evaluation rounds. The efficacy of the two algorithms over existing approaches (e.g., Gradient Descent with one/two-point feedback, HOO, and UCB) are demonstrated using numerical experiments as well as some illustrations.

**Audience:**

Yes

**Broader Impact Concerns:**

I believe there is no concern associated to the developments and results reported in the paper.

**Claims And Evidence:**

Yes

**Requested Changes:**

See the comments above.

**Strengths And Weaknesses:**

The paper investigates an interesting problem, which is of high relevant to TMLR. I believe it renders attractive to audience from both optimization and (bandit) learning communities. Further, as the studied resource allocation problem admits a generic form, despite its framing for the setting of programmatic advertising optimization, I believe the algorithms presented in the paper will find use in other application areas in engineering where similar resource allocation problems would naturally emerge.

The paper is written very well, despite the typos I spotted and some places that could be further improved; see below. Further, it has a clear presentation and is overall a nice read. A positive aspect here is to have included in the introduction a well-elaborated motivating example, which fits well the importance of the setting and the gap in the literature.

In terms of technical strengths, one, in my view, is to have considered a regret definition in terms of function evaluations, as opposed to the classical ones in the literature. Also, another one is the consideration that only feasible points could be queried. These make the presented algorithms more faithful to practical scenarios.

The two proposed algorithms are designed and analyzed under a set of assumptions on the objective function (Assumption 1) commonly considered in the most of literature on bandit convex optimization. A positive aspect of them is that the assumptions, unless I miss something, are only there to ensure the reported regret bounds, and the algorithms do not rely on the knowledge of strong convexity or smoothness moduli. This might be worth highlighting. Further, it appears that the algorithms could be still used even if some of assumptions (e.g., strong convexity) are violated; namely, even if the analysis could be open under such relaxations, the empirical performance could nonetheless be sound. However, this is neither investigated nor discussed in the paper.

One drawback here is that when comparing to the regret bounds of approaches such as UCB + discretization, it is not highlighted that they may not rely on such assumptions. It appears to me that the proposal by Combes and Proutiere (2014), although suffering from a larger regret when the dimension $d$ grows, would rely on arguably milder assumptions of (mostly) implying unimodality of the decision space. I urge the authors to precisely discuss this matter in the paper.

Other major drawbacks and comments are listed below:

1. It is assumed that the optimal allocation resides in the interior of the feasible sets. To what extent is this assumption restrictive in practice? I wonder whether relaxing this assumption would lead to bad empirical performance or maybe it is there merely for the regret analysis.

2. The reported regret bounds do not elaborate well on the dependence of the bound on relevant parameters of interest, especially $a$, $\beta$, $\sigma$ etc. Also, I found it difficult to get these from the more detailed bounds in the appendix. I urge the authors to further elaborate on this and potentially discuss such dependence. I understand that the final bound could be rather ugly, as a function of all involved quantities. Yet, discussing such dependencies would be very helpful.

3. Continuing the previous point, a reader might be curious on how the derived regret bounds stand with respect to other bounds in terms of dependence on $\beta$ and $a$ (of course to the extent that comparison would make sense).

4. I appreciate the comparison made with UCB in some respects. However, in the end, it is not clear for the cases of small $d$ (say, at most 4) which of UCB or FDS-Seq is advised theoretically and empirically. In the case of $d=2$, UCB achieves a better regret bound than FDS variants. Also, the experimental result reported in Section 4 suggests that it could be a viable solution in small $d$.

5. While most of the paper is written very well, the author(s) have been less careful in writing Subsection 3.2. I spotted many notation inconsistencies there and some typos. Below I listed what I spotted, but I’d strongly recommend the authors to carefully revise this part.

6. Some details of “Illustration” (Section 2.2) could be moved to the appendix (e.g., the long paragraph on how HOO is adapted). This could be more relevant if some space in the main text is needed to accommodate the requested revisions.

Further detailed comments:

- In algorithms it is mentioned $\theta \leq 1$ whereas in the text $\theta<1$ required.

- p. 5: $D_k$ must be $\mathbb D_k$

- p. 6: $\gamma$ is used both as a parameter in $w_i(x)$ and to denote a growth factor. Although clear, consider using different symbols.

- p. 6: The statement “and the edges to linear paths along which …” appears which unclear to me. Could you revise it and ensure, e.g., a verb is not missing?

- p. 8: FDS-Test => FDS-Seq

- p. 8: In Theorem 1, it appears that a right parenthesis is missing in the first term. Please check it carefully. Also, it looks like plugging-in the definition of $\rho$ (and some simplification) could make the bound more interpretable.

- p. 9: In “the gap $f(x_k) – f(x_k - \alpha_k v)$”, did you mean $f(x_k) – f(x_k +\alpha_k v)$?

- Assumption 4: In view of the $\beta$-smooth assumption, assuming a bound on $\nabla f$ appears redundant to me, if we have a bounded set $\mathcal D$.

- p. 10: In “This property is not sufficient in the unconstrained case”, did you mean “the constrained case”?

- Why “FDS-Plan and gradient descent show roughly the same regret”?

- Some details appear missing in relation to the experimental results. E.g., what is it the error bars indicate.

- Theorem 4 in appendix: You assume that the region where $f(x) < f(x_0)$ is convex. But isn’t it simply an implication of convexity of $f$? I am referring to the property that all $c$-level sets of a convex function $f$ are convex sets.

Potential Typos/Grammar Mistakes:

- p. 3: of and increased regret rate => … an increased …

- p. 3: The algorithm that they propose relies on => The algorithms they propose rely on (Note: you are referring to two works, hence algorithms.)

- p. 3: Flaxman et al. (2004) later considers => Remove “later” as the preceding sentence cites a work from 2017.

- p. 4: allows to understand => allows for understanding OR allows us to understand

- p. 5: we replace … by => … with

- p. 6: In “Finally, … of FDS-Plan (respectively FDS-Seq) occurred are …”, it seems that “and” could be used in lieu of “respectively”.

- p. 6: two algorithm => two algorithms

- p. 9, Lemma 2: Assumption 1 and 2 => Assumptions

- p. 9, Lemma 2: a unsuccessful iteration then => an unsuccessful iteration, then

- p. 10: i-th => $i$-th

- p. 10: “… are $x$ are denoted …”: Please carefully check the sentence.

- p. 10, Assumption 5, Lemma 10 (and potentially elsewhere): “{” and “}” are missing.

- p. 10: in the following, => in the following.

- p. 9 (and elsewhere): $x_*$ => $x_\star$ (to be consistent with earlier definitions).

- p. 11: $R^2$ => $\mathbb R^2$

- p. 11: $\mathbb R_d$ => $\mathbb R^d$

- p. 11: What is $G_k$? What is $\Pi$ when you wrote “($2\Pi/3$ apart)”?

- p. 12: … they rely on => it relies on

- p. 20: Make correction to $\hat f(x_k\alpha_kv)$.

- p. 18, Lemma 5: extra parentheses in the definition of $C_1$.

- p. 22, Lemma 16: $log$ => $\log$

- p. 24: Carefully check the exponent $-32/3\times 3/2$

---

> ### Author Response · Authors · 2024-03-05
> **Response to Review**
>
> We would like to thank the reviewer for their careful reading of the manuscript and very detailed review. We only cover the most important points in the response, but we have taken into account all the remarks in preparing the updated version of the manuscript.
>
> 1. Optimal allocation in the interior: Assuming optimal allocation in the feasible set’s interior is indeed crucial for analysis, but we believe that the existence of such a degenerate optimum is not  harmful in practice, as supported by simulations (see Appendix E of the updated version).
>
> 2, 3. Dependence on regularity parameters: The dependence on regularity parameters $a$ and $\beta$ is not highlighted because of its rather complex form. The dependence of the regret of UCB on a discretization of the domain is linear in the smoothness parameter $\beta$ (see Combes & Proutiere (2014)), and it would hence be natural to expect the same kind of rate. However, even in the simpler case of Theorem 1, it is obvious that the regret rate depends in a complex way on both $\beta$ and the ratio $\beta/a$. We understand the reviewer’s concern but it seems that there is no simple form of dependence that can be usefully highlighted in the bounds.
>
> 4. Comparison to UCB (required assumptions and performance): We agree that in small dimensions, UCB might be the simpler solution with good performance rates. In dimension $d=2$, the asymptotic behavior of the regret of UCB is of order $\sqrt{T}$, like that of HOO. The asymptotic behavior will degrade in higher dimensions, reaching a rate of $T^{d/(d+2)}$ for the optimal discretization. The assumptions required to obtain this order of regret for UCB are indeed slightly less constraining than Assumption 1 because they only require a quadratic upper and lower bound on the function locally near the optimum, whereas we require conditions that hold uniformly on the domain. However, it is likely that it is an artifact of the analysis. It is true that Combes & Proutiere (2014) also analyze the unimodal case with even weaker regularity assumptions but these lead to weaker regret guarantees.

---

### Review · Reviewer_GNrJ · 2024-02-21

**Summary Of Contributions:**

The paper studies direct search methods for blind resource allocation. In particular, the authors study a repeated problem in which at each time step an agent chooses a feasible allocation and then observe a stochastic reward. This framework models the problem of allocating budget to advertising campaigns, but it is applicable to more general problems. The paper provides a direct search method that provides regret $T^{2/3}$ under linear constraint. Finally, the paper provides a more practical algorithm with better empirical performances.

**Audience:**

Yes

**Claims And Evidence:**

Yes

**Requested Changes:**

I suggest to better clarify the generality of the model and that budget allocation is only an (interesting and important) application.

**Strengths And Weaknesses:**

Strengths:
The paper is study an interesting problem providing efficient solutions. Moreover, the paper shows that other algorithms (that were designed for similar problems) are less effective also from a practical point of view.

Weakness:
The work is rather incremental, and it relies heavily on techniques introduced by previous work.
The setting under study is not clearly presented. While the motivating application is budget allocation, the setting is quite more general and should be introduced in its generality.

---

> ### Author Response · Authors · 2024-03-05
> **Response to Review**
>
> We thank the reviewer for their comments on the manuscript. We have revised the introduction to emphasize that the problem we tackle extends beyond blind resource allocation. However, our primary focus remains on offering a practical algorithm for blind resource allocation with a reasonable regret rate, knowing that for simpler constraints (in particular box constraints), there already exists practical algorithms (HOO, UCB on grid) with good regret rates.

---

### Author Response · Authors · 2024-03-05
**General comment**

We thank the reviewers for their feedback on the manuscript. Based on these, we are proposing a revised version of the manuscript, where differences to the original submission are highlighted in blue. We cover the specific points raised by the reviewers in the individual answers to each review.

---

### Decision · Action_Editor_AFUT · 2024-03-26

**Recommendation:** Accept with minor revision

**Comment:**

This submission delves into the domain of programmatic advertising optimization, focusing on the sequential budget allocation across various resources under linear constraints and noisy evaluations. The paper proposes novel direct search algorithms that are analyzed for regret in both deterministic and noisy conditions. It presents an extensive theoretical analysis along with empirical evaluations to demonstrate the efficacy of the proposed methods.

The paper's strengths include its innovative algorithmic solutions for resource allocation challenges, thorough theoretical insights, and convincing empirical evidence. On the other hand, two of the reviewers have both questioned the originality of the algorithm design and its analytical aspects. Additionally, the reviewers have noted the lack of convincing motivation behind some modeling choices and questioned the scalability and practical applicability of the proposed methods. Therefore, I suggest that the authors revise their paper to highlight the novelty of their algorithms and their practical value.

**Audience:**

Yes

**Claims And Evidence:**

Yes